# Amplification of downstream flood stage due to damming of fine-grained rivers

Hongbo Ma [1✉], Jeffrey A. Nittrouer [2✉], Xudong Fu[3✉], Gary Parker[4,5], Yuanfeng Zhang [6], Yuanjian Wang[6], Yanjun Wang[3], Michael P. Lamb [7], Julia Cisneros [4], Jim Best [4,5,8,9], Daniel R. Parsons [10] & Baosheng Wu[3]

River dams provide many benefits, including flood control. However, due to constantly evolving channel morphology, downstream conveyance of floodwaters following dam closure is difficult to predict. Here, we test the hypothesis that the incised, enlarged channel downstream of dams provides enhanced water conveyance, using a case study from the lower Yellow River, China. We find that, although flood stage is lowered for small floods, counterintuitively, flood stage downstream of a dam can be amplified for moderate and large floods. This arises because bed incision is accompanied by sediment coarsening, which facilitates development of large dunes that increase flow resistance and reduce velocity relative to pre-dam conditions. Our findings indicate the underlying mechanism for such flood amplification may occur in >80% of fine-grained rivers, and suggest the need to reconsider flood control strategies in such rivers worldwide.

[1] Department of Civil and Environmental Engineering, University of California at Irvine, Irvine, CA, USA. [2] Department of Geosciences, Texas Tech University, Lubbock, TX, USA. [3] State Key Laboratory of Hydroscience and Engineering, Tsinghua University, Beijing, China. [4] Department of Geology, University of Illinois, Urbana-Champaign, IL, USA. [5] Department of Civil and Environmental Engineering, Ven Te Chow Hydrosystems Laboratory, University of Illinois, Urbana-Champaign, IL, USA. [6] Yellow River Institute of Hydraulic Research, Zhengzhou, Henan, China. [7] Division of Geological and Planetary Sciences, California Institute of Technology, Pasadena, CA, USA. [8] Department of Geography and Geographic Information Science, University of Illinois, Urbana-Champaign, IL, USA. [9] Department of Mechanical Science and Engineering, University of Illinois, Urbana-Champaign, IL, USA. [10] Energy and Environment Institute, University of Hull, Hull, UK. ✉email: bigmatton@gmail.com; jeffrey.nittrouer@ttu.edu; xdfu@tsinghua.edu.cn

Since the beginning of human civilization, society has maintained a history of life alongside rivers[1], manipulating them through installation of dams and reservoirs. This has arisen to control floods, generate energy (e.g., mills, hydropower), facilitate navigation, and secure water resources[2]. Although there are currently >45,000 large dams (height > 15 m) distributed over 140 countries[3], the geomorphic disturbances wrought by dams are not fully understood[1,4–6]. Moreover, flood-risk evaluations of dammed rivers are difficult to perform, since the fluvial channel is constantly evolving[5,7,8]. For the same reason, despite studies showing river engineering (e.g., built levees, dikes, and other impedances to natural flow) can have adverse impacts, including a greater flood risk, some of the underlying mechanisms are still in doubt[8–11]. This is a non-trivial problem because river flooding is a devastating natural hazard, with the size and frequency of events predicted to increase due to climate change (e.g., extreme precipitation, changing landscape and vegetation coverage[1,12–17]). It is thus increasingly necessary to evaluate the safety and stability of river channels subject to extreme floods, and assess how to mitigate risk.

Flood risk is typically evaluated by the flood-water stage: $Z = Z_b + H$ [L], where $Z_b$ is channel bed elevation [L] and $H$ is water depth [L] (Supplementary Fig. S1a). If $Z$ surpasses the elevation of a levee top, this may lead to the levee being breached, which for large rivers could unleash a potentially devastating flood. One of the benefits of damming river channels is that the erosional capability of sediment-depleted flow works to lower $Z_b$ downstream of dams[5] and thus may decrease water stage and flood risk[6,7]. Along with bed degradation, selective entrainment drives bed sediment coarsening, whereby finer material is disproportionally entrained[18], and coarser material gradually dominates the bed grain-size distribution. In turn, resistance to degradation improves over time, and the bed may eventually reach a quasi-equilibrium elevation[4,18–20]. In addition to deepening, c. 46% of dam-impacted rivers exhibit channel widening[4,21] due to bank collapse (Supplementary Fig. S1b). It has also been documented that 26% of deepening channels develop a narrower width[4], whereby the abandoned channel bed reverts to floodplain (Supplementary Fig. S1c). Thus, the cross-sectional area of the channel enlarges by both deepening and widening[21] (Supplementary Fig. S1c).

The hypothesis that an incised and enlarged channel downstream of a dam results in lower flood stage, and therefore reduces downstream flood risk, has been utilized to justify the development of large dams[22,23]. However, the fact that channel enlargement can trigger changes in flow resistance between pre- and post-dam periods is often not considered, but yet can also impact the flood-stage elevation. For instance, the water depth ($H$) for a given flood discharge ($q_w$) can be formulated in terms of a resistance coefficient ($C_f$):

$$H = (C_f/gS)^{1/3} q_w^{2/3}, \qquad (1)$$

where $S$ is the water surface slope [−], $g$ is gravitational acceleration [LT$^{-2}$] and $q_w$ is flow discharge per-unit width [L$^2$T$^{-1}$]. Equation 1 has a mechanistic derivation[24–26] where $C_f$ is highly dependent on bed roughness; specifically, bedform size plays a key role[27–30]. Herein we test the hypothesis that the installation of a dam on a fine-grained river (silt-bedded and sand-bedded rivers with median bed grain size <2 mm) systematically impacts the resistance relation (i.e., $C_f$ in Eq. 1) by coarsening bed sediment and changing bedform size and geometry. Thus, rather than the intended consequence of reducing flood stage due to bed incision downstream of a dam, heightened flow resistance amplifies flow depth and stage and increases flood risks.

Bed coarsening after dam construction is a common phenomenon, and found in multiple dammed large rivers worldwide, e.g., the Missouri-Mississippi River[4,19], Colorado River[4], and Yellow River[18]. After dam closure, a wave of bed-coarsening is initiated at the dam site and propagates downstream. The typical propagation rates of this coarsening wave are estimated to be 10–70 km/yr[4,18,19]. The time-dependent nature of this process allows for testing of rivers currently in a state of transition. The study site considered herein is the Xiaolangdi Dam (XLD) on the lower Yellow River (LYR), China (Fig. 1), where rapid morphological changes since dam closure (1999) have been documented[22,31], primarily due to the fine bed material size (median grain size < 0.150 mm i.e., silt to fine sand) (Fig. 1b), and high sediment transport rates[32,33]. The LYR provides an ideal case study, as it is currently in the process of transitioning in its grain-size response to damming, and its extensive records provide both pre- and post-dam data that include bed material size, channel geometries, and flow depth, stage and discharge—data that are not readily available for most of the world's large rivers. These data enable estimates of changes in bed roughness and flow resistance, and their impact on river stage (see Supplementary Table S1). We augment these data with multibeam and parametric echo sounding bathymetric surveys collected at the upper- and lowermost locations on the LYR, which provide direct and contrasting evidence for changes in bed roughness and flow resistance. Lastly, using a global database, we derive a threshold that accounts for water stage amplification, extending to other fine-grained rivers worldwide.

## Results

**Bed material coarsening due to dam installation.** The median bed material grain size at seven gauging stations on the LYR (Fig. 1a) was measured from samples collected annually over a 50-yr period beginning in 1965 (i.e., 34 years pre-dam, and 16 years post-dam; see Supplementary Table S1 and Text "Hydraulic, cross-section and bed material database" in Methods). Pre-dam, the variation in median grain-size ranged from 0.04 to 0.15 mm (Fig. 1c), but after completion of the XLD, bed material size coarsened rapidly at all stations (Fig. 1c). At the uppermost station, Huayuankou (~150 km downstream of the XLD), the measured median bed grain size increased to ~0.3 mm. Grain size becomes finer downstream: at Jiahetan, 125 km from Huayuankou, median grain size is ~0.15 mm, and stations downstream have beds finer than 0.15 mm (Fig. 1c, d).

We focused on thalweg samples for the reach from upstream of Huayuankou to downstream of Gaocun (see Fig. 1b and Text "Refined bed material surveys" in Methods). The location where thalweg grain size is greater than 0.15 mm is between the Jiaoyuan Floating Bridge (~290 km from the XLD dam) and Gaocun station (~340 km from the XLD). The reach upstream of this is hereafter termed the "impacted reach" (i.e, XLD to Jiaoyuan Floating Bridge), and also corresponds to where significant changes in pre- and post-dam river morphology and flood characteristics are expected. The reach downstream of Gaocun remains within the range of pre-dam grain-size variation and is thus termed the "non-impacted reach", where pre- and post-dam river morphology and flood characteristics are expected to be similar. Therefore Huayuankou (the uppermost gauging station of LYR), and Lijin (the lowermost gauging station of LYR; Fig. 1a) represent impacted and non-impacted reaches respectively, with the median grain size in the most recent surveys being 0.3 mm (2015; Huayuankou) and 0.09 mm (2016; Lijin).

**Bedform geometry changes in response to bed coarsening.** The conditions that control bedform geometry in large, fine-grained

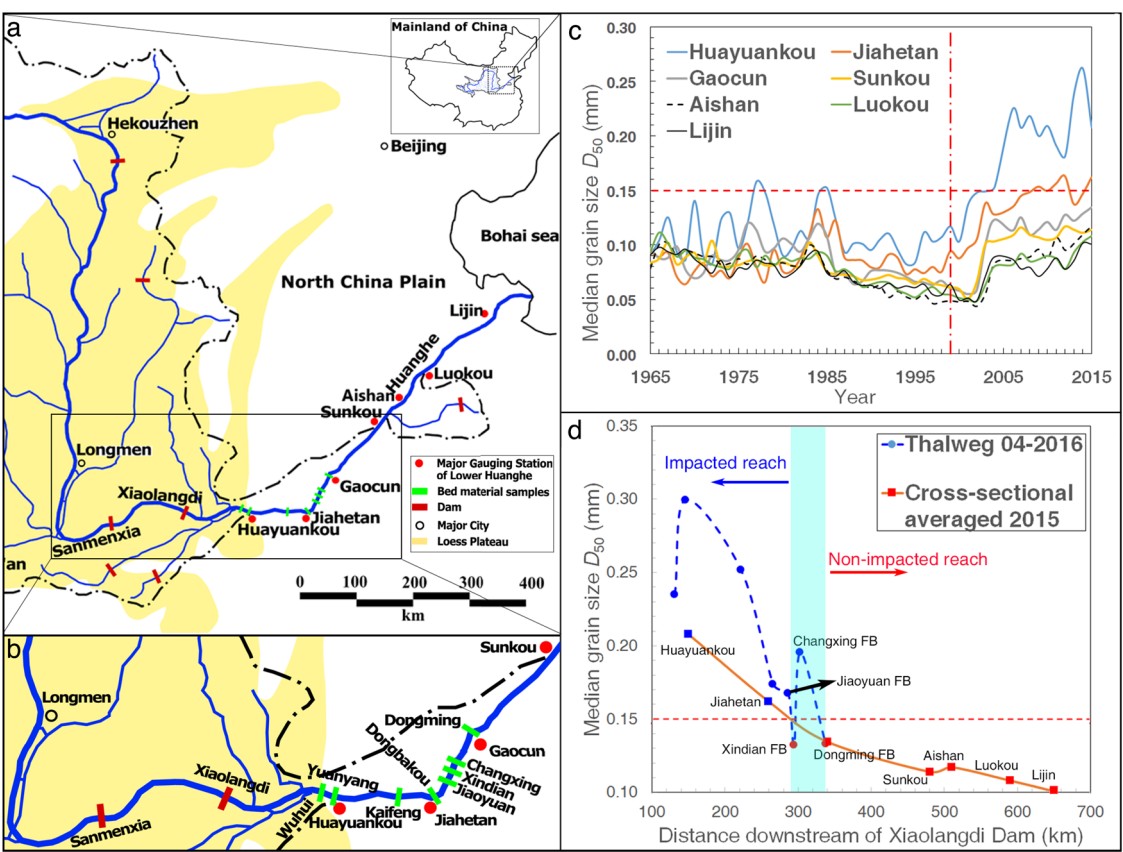

**Fig. 1 Maps of the lower Yellow River (Huanghe), China, and trends in spatio-temporal evolution of bed material grain size. a** Map of the lower Yellow River; **b** Detailed view of box shown in (**a**) illustrating locations for sampling of thalweg bed material; **c** Cross-sectionally averaged bed material grain size, surveyed at the seven major gauging stations, over a 50-yr period (see locations in (**a**). Vertical dot-dashed line demarcates the time when the Xiaolangdi Dam became operative. Horizontal dashed line demarcates the upper bound of bed material (0.15 mm) under pre-damming conditions; **d** Bed material grain size for the thalweg of the upper reach of the LYR. The transition between the non-impacted and impacted reaches of the LYR (blue shading) is located between Jiaoyuan Floating Bridge and the Gaocun gauge station; red markers indicate where median grain-size drops below 0.15 mm.

rivers represent an important, yet still unresolved problem[28]. Observations from both natural channels and laboratory experiments indicate that as flow strength increases from the initiation of grain mobility, a lower-stage plane bed or rippled bed (characterized by relatively little sediment transport) develops; additional bed shear stress produces dunes[34], and eventually, with sufficient stress, the bed transitions from a dune field (with maximum bedform heights of ~1/3 of the flow depth) to an upper stage plane bed, passing through a stage of dune washout (i.e., bedform height much <1/10 the flow depth)[27,29,30,35–37]. Correspondingly, the roughness created by bedforms grows from a minimum to a maximum, before again decreasing to a minimum[38]. Roughness as a function of flow discharge can thus be approximated by a hump-shaped (e.g., parabolic) function of an appropriate transport parameter, which herein is the suspension number, $u^*/v_s$ (where $u^*$ and $v_s$ are the shear velocity and sediment settling velocity, respectively [$LT^{-1}$]).

A global database (see Supplementary Table S2) including laboratory and riverine bedform data[39,40] was used to develop a quantitative relation between relative bedform height and suspension number (Fig. 2), excluding cases of high Froude number (Fr; i.e., Fr > 0.8) for reasons discussed below. This quantitative relation shows convincing predictability compared to existing models (Supplementary Fig. S2 and Supplementary Table S4). The database has a grain size range 0.13–36 mm, with a focus on finer-grained channels ($D_{50} < 2$ mm), which comprise ~80% of the data. We assess that bed coarsening after damming leads to a drop in the suspension number ($u^*/v_s$), based on the

facts that grain size usually responds first to the disturbance to the fluvial system[41,42] and that the settling velocity ($v_s$) is sensitive to grain size changes ($v_s \propto D_{50}^{2\sim0.5}$)[43]. Therefore, for fine-grained rivers, bed sediment size is a more important factor than potential change in shear velocity after damming.

The relation developed from the analysis of bedform height and suspension number (Fig. 2) predicts that the bed grain-size differences measured at Huayuankou and Lijin should correspond to different bedform geometries, and in turn, variable bed roughness. For example, for the bed at Lijin (non-impacted), the suspension number $u^*/v_s \sim 10$ is indicative of an upper stage plane bed, whereas at Huayuankou, this ratio is $u^*/v_s \sim 1–2$ and predicts the occurrence of dunes (Fig. 2). In this example, the suspension number drops rapidly at Huayuankou as the bed coarsens post-damming. Field observations also show the decrease of suspension number post-dam (Supplementary Table S3).

More than one physical process can lead to the transition between a dune field and an upper stage plane bed. The high suspension number, employed above, represents the effect of the high fraction of suspended to total load on the washout of the large dunes, thus leading to an upper stage plane bed. In addition, the high Froude number (Fr) may also result in such a transition, which represents the effect of the interaction between surface waves and bedform geometry. Both factors exert significant controls on bedform shapes[40,44,45]. Recent studies, including flume experiments[27,35,46], field observations[47], and theoretical analyses[48], demonstrate that sediment suspension represented by

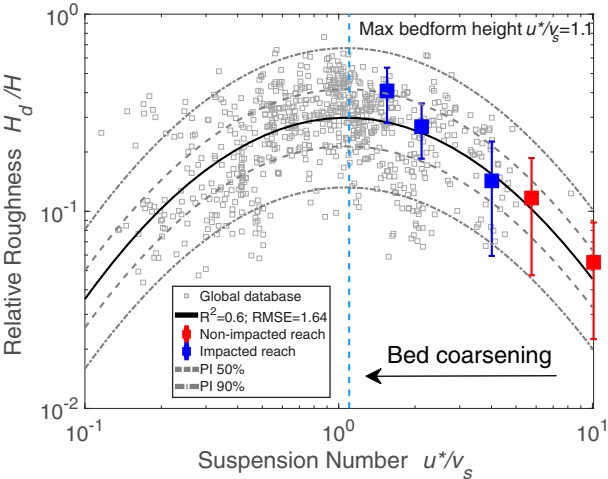

**Fig. 2 Relation between relative bedform (roughness) height and suspension number.** A comprehensive database (gray squares), including both laboratory and field data, is compiled to determine the hump-shaped relation (solid line) between relative bedform height and suspension number, where $H_d$ is dune height [L], $H$ is water depth [L], $u^*$ is shear velocity [$LT^{-1}$], and $v_s$ is the sediment settling velocity [$LT^{-1}$]. Although such hump-shaped relations are well recognized[27,29,30], a well-tested quantitative relation remains lacking heretofore. The proposed relation, based on a parabolic equation in logarithmic space, $\log_{10} Y = -0.87(\log_{10} X - 0.03)^2 - 0.53$, shows a good fit and covers a wide range of laboratory and field data. The comparison among the proposed relation, the compiled database and other relations are given in Supplementary Fig. S2. Field observations of bedform heights in the dam-impacted (blue squares) and non-impacted reach (red squares) of the lower Yellow River (LYR) agree with the proposed relation very well. The squares represent the mean value of relative bedform height and error bars represent one standard deviation. In particular, the observed data from the non-impacted reach of the LYR quantify a previously-untested region of the proposed relation. As shown in Supplementary Table S3, the difference in suspension number (and thus bedform geometry) in the impacted and non-impacted reaches results mainly from bed coarsening. The hump-shaped relation indicates that if the bankfull suspension number is greater than a suspension number corresponding with the dune-size maximum (~1.1 via present model; vertical blue dashed line), bed coarsening induced by damming on the river will result in larger dunes during floods. Dashes and dashed-dotted lines represent the 50% and 90% prediction intervals (PI) of the present model, respectively.

the suspension number in fine-grained rivers is the predominant control on bedform geometry and their state transition, because these systems typically maintain low Fr (Fr ≪ 0.8).

To further test the assessment that bedform morphology varies significantly with suspension number in the dammed river, we measured bedform geometry at Huayuankou (Figs. 3a, S3) and Lijin (Figs. 3b, S3) (see Text "Channel bathymetric surveys" in Methods). While the channel bed at Lijin (non-impacted) is relatively flat (maintaining a bedform height relative to flow depth much <10%), the bed at Huayuankou (impacted) is characterized by large dunes possessing relative dune heights and aspect ratios (length-to-height) similar to typical sand-bedded rivers (Figs. 3c, S3e), such as the Mississippi River[32,49]. Observations of bedform geometry from Lijin and Huayuankou agree with the proposed relation between bedform height and suspension number (Fig. 2). Furthermore, data from Lijin suggest that the transition from a large, sand-bedded dune field to subtle, low-relief bedforms is gradual (i.e., $4 < u^*/v_s < 10$; Fig. 2), challenging conventional views on bedform shape change (an

abrupt change; see Supplementary Fig. S2) between silt- and sand-bedded rivers. It is possible that the bedforms with large aspect ratio in silt- and sand-bedded rivers maintain very long wavelengths (Fig. 3) that cannot be captured by experimental studies. The development of an extensive dune field at Huayuankou, compared to the relatively flat bed at Lijin, therefore supports the hypothesis that dam-induced bed coarsening can play an important role in establishing large dune fields, which ultimately change the roughness structure and alter flow resistance and depth.

**Changes in the resistance relation and flood stage after damming.** As the resistance relation (Eq. 1) pertains to steady and uniform (normal) flow[50,51], a dataset of normal flow and equilibrium sediment transport conditions was selected from hydraulic data collected by local hydrological stations (Supplementary Table S1 and Text "Equilibrium database on hydraulic and sediment transport" in Methods), and used to establish a reference value for flow resistance (Fig. 4a). The change in the resistance coefficient ($C_f$) between pre- and post-dam conditions at Huayuankou (Fig. 4a) shows a mean post-damming $C_f$ value of 0.0046 (reaching as high as 0.01), which is three-fold greater than the pre-dam value of 0.0015. Notably, the post-damming $C_f$ at Huayuankou is comparable to values for typical sand-bedded rivers[52]. Uncertainty in the resistance coefficient may result from the multi-threaded channel pattern at Huayuankou. When examined together, the pre- and post-damming resistance coefficients agree with the hump-shaped curve resulting from the compiled global database (Fig. 4b). It is prudent to note that, due to a lack of data in the range of suspension number from 3 to 10, the empirical relation developed from the global database could not claim applicability to this range until data from the LYR are used to confirm it. Independently verified data from the LYR thus quantify a previously-untested region of the relation up to a suspension number of 10. There is a strong similarity in the hump-shaped relations for relative bedform height (Fig. 2) and resistance coefficient (Fig. 4b). The threshold suspension number corresponding to maximum relative dune height (1.1) and maximum resistance (0.9) are also very close, again supporting the hypothesis that the change in bedform geometry impacts channel resistance significantly.

The mean values of $C_f$ are used to represent the average resistance conditions for pre- and post- damming. Substituting mean $C_f$ values into Eq. 1 and using the average channel slope at Huayuankou ($S = 3 \times 10^{-4}$), the theoretical predictions agree well with field measurements of water depth at Huayuankou (Fig. 4c). Importantly, the post-damming water depth for a given flood discharge (per-unit-width) $q_w$, increases on average by a factor of 1.4. In the extreme scenario (i.e., $C_f = 0.01$), the increase is a factor of 2.0, which is an occurrence that will become more frequent as the bed continues to coarsen (i.e., relation in Fig. 4b). Meanwhile, the water depth in the non-impacted reach of the LYR (Lijin), where grain size remains similar to the pre-dam condition, shows no change (Supplementary Fig. S4). Bed coarsening thus coincides with increasing dune size and roughness, which increases channel resistance. In turn, water depth is elevated for the same per-unit-width flood discharge.

We use a hydraulic framework accounting for water depth $H$ and cross-sectional area to assess how $Z_b$ and channel width $W$ [L] affect flood stage (i.e., $Z = Z_b + H$, see Text "Hydraulic prediction for the flow stage based on observed cross-sections" in Methods). For instance, the channel bed at Huayuankou degraded by 3.44 m after construction of the XLD ($Z_{b-post} = Z_{b-pre} - 3.44$), as can be determined by comparing the cross-section data from 2015 to 1981 (Fig. 5a). Additionally, more

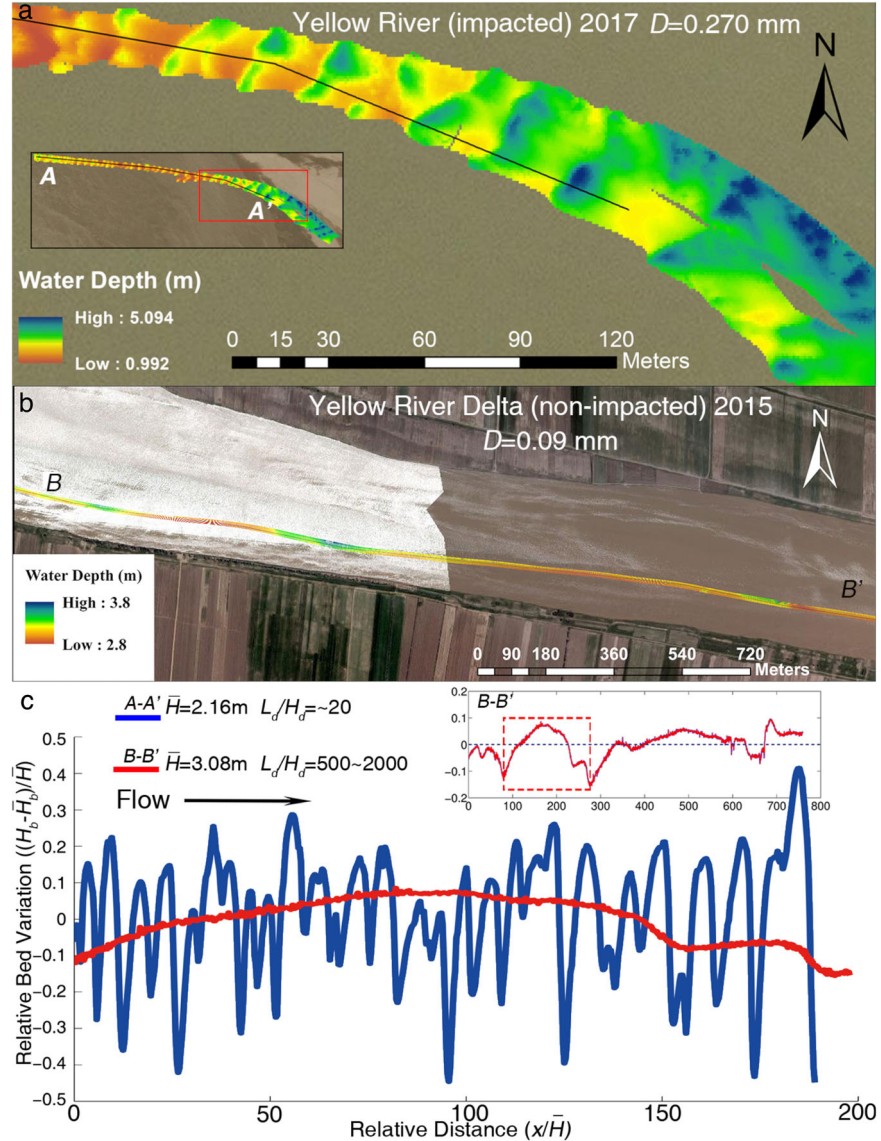

**Fig. 3 Bedform geometry for the present-day Yellow River. a** Multibeam echosounder (MBES) map of periodic, high-relief dunes ($L_d/H_d \sim 20$ and $H_d/\bar{H} \sim 40.81\% \pm 12.72\%$, where $L_d$ and $H_d$ are wavelength and wave height of bedforms [m], respectively and $\bar{H}$ is the average water depth [m]). These bedforms are found in abundance at Huayuankou, just downstream of Xiaolangdi Dam (see Supplementary Fig. S3 and Supplementary Table S3); **b** Low-relief bedforms ($L_d/H_d \sim 500$–2000 and $H_d/\bar{H} \sim 5.51\% \pm 3.26\%$) near Lijin, where bed sediment size is not impacted by the XLD dam; **c** Longitudinal profiles through the bedforms from the two studied reaches derived from the MBES maps. $H_b$ and $\bar{H}_b$ represent the bed elevation along the transect and average bed elevation of the survey area, respectively. The fine-grained channel of the non-impacted downstream reach is characterized by low-relief bedforms, while the channel bed of the dam-impacted reach possesses high-relief dunes typical of sand-bedded rivers. The presence of low-relief bedforms is the primary reason for a lower flow resistance.

than 20 years of cross-section data (1981–1990 for pre-dam and 2006–2015 for post-dam) were collected to assess stage change pre- and post-damming. Flood stages of each event were computed based on measured cross-sectional shape for each year, using the theoretical resistance relation (Text "Hydraulic prediction for the flow stage based on observed cross-sections" in Methods). The model has no adjustable parameters, nor are $Q_w$ or $Z$ used to calibrate any parameter. Figure 5b shows theoretical predictions and observed data for different flood flow discharges (see Fig. S6 for a one-by-one comparison). The general trends of the $Q_w$–$Z$ relations are shown as pre-dam (red) and post-dam (blue). The post-dam $Q_w$–$Z$ relation without any effects of stage amplification produced by resistance enhancement is also indicated (pink). Cross-section variations are included via the flood-stage confidence intervals corresponding to flood discharge.

If it is assumed that the relation $H(q_w)$ is unchanged (i.e., no flood-stage amplification effect), then bed degradation would produce ~3.0 m lowering of flood stage (i.e., pink line in Fig. 5b). However, the field data (blue) indicate that flood stage for the post-dam condition fell little for small flood events (e.g., pre-dam bankfull discharge 4000 m³ s⁻¹). Furthermore, for moderate and large floods, the post-damming stage is amplified, surpassing the pre-damming stage at a discharge of $Q_w = 6099$ m³ s⁻¹ (95% CI [4847, 7889]), which represents a flood event less than the 5-yr recurrence interval flood (10,000 m³ s⁻¹) at Huayuankou[53].

Channel bed degradation, as a long-term accumulative effect induced by dams, lowers flood stage by a slowly-varying value with respect to water discharge (blue line in Fig. 5c), whereas enhanced resistance works as a multiplier (Eq. 1) to amplify flood stage (e.g., red line in Fig. 5c). Hence, as the net result of both

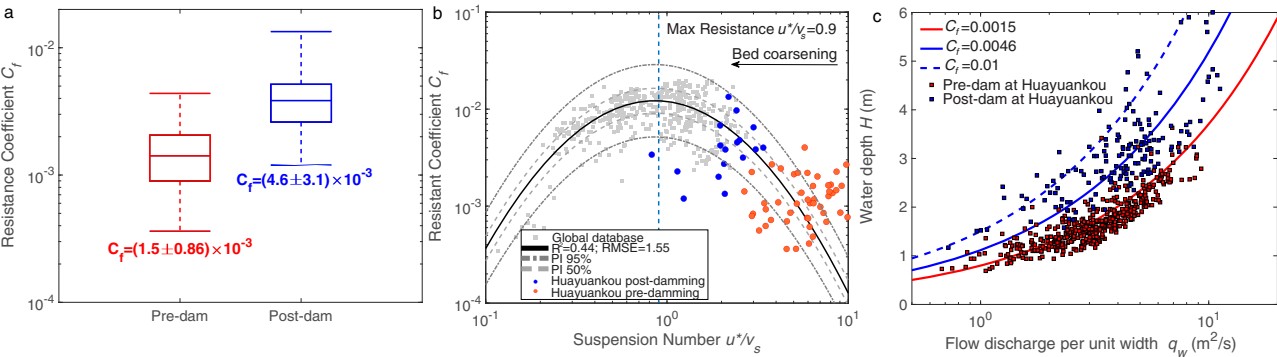

**Fig. 4 Adjustment of flow resistance coefficient $C_f$ at the impacted reach of the Yellow River, before and after construction of the XLD dam, and its subsequent impact on flood-water depth.** **a** The average value of $C_f$ in the post-damming bed roughness state is three times that for the typical pre-damming state at Huayuankou, the impacted reach of the Yellow River. Elements of the box-plot include: center line, mean value; box limits, upper and lower quartiles; whiskers, max and min values; **b** Relation between resistance coefficient and suspension number. The resistance coefficient at Huayuankou under pre- and post-damming conditions agrees well with the proposed hump-shaped relation, e.g., $\log_{10}Y = -1.75(\log_{10}X + 0.066)^2 - 1.91$; **c** Comparison of $H\text{-}q_w$ relations at the impacted reach of the Yellow River before and after dam construction. The figure illustrates the theoretical relation $H = (C_f/gS)^{1/3}q_w^{2/3}$ with $C_f$ values inferred in (**a**), and independent field data collected at Huayuankou, for both pre- and post- dam conditions. The blue solid line, using averaged post-damming $C_f = 0.0046$, shows $H$ is 1.4 times greater than pre-damming; blue dashed line, using the maximum post-damming $C_f = 0.01$, illustrates $H$ is 2.0 times greater than pre-damming. Comparison of $H\text{-}q_w$ relations at the non-impacted reach of the Yellow River (Lijin) before and after dam construction is shown in Supplementary Fig. S4.

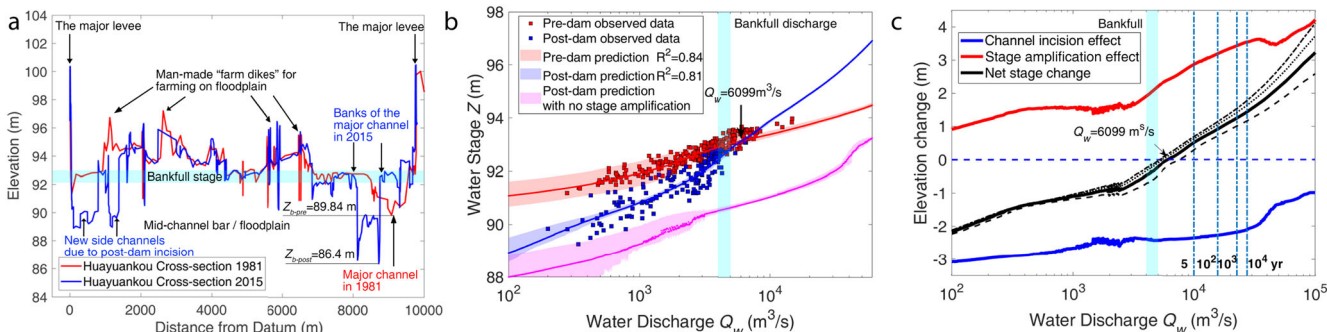

**Fig. 5 Evaluation and comparison of the overall flood stage for pre-dam and post-dam conditions.** **a** Cross-sectional profiles for pre-dam and post-dam conditions at the Huayuankou gauging station; **b** Pre- and post- dam flood stage and flood discharge. The red solid line is the theoretical prediction for the pre-dam flood stage based on observed cross-sections and Eq. 1 with $C_f$ from Fig. 4b (see Text "Hydraulic prediction for the flow stage based on observed cross-sections" in Methods), and the red shaded area indicates the 95% confidence interval of the prediction uncertainty induced by the pre-dam cross-section variation. The blue solid line is the theoretical prediction for the post-dam flood stage, with the blue shaded area indicating the 95% confidence interval of the prediction uncertainty induced by the post-dam cross-section variation. The pink solid line is the theoretical prediction for the post-dam flood stage based on the post-dam cross-sections (including considerations of channel widening and bed incision) without consideration of the stage amplification effect (grain size and bedform size unchanged). **c** Water stage elevation changes produced by various effects. The blue solid line represents the stage elevation change induced solely by channel incision (enlargement) without consideration of the stage amplification effect. The red solid line represents the stage amplification effect. The black solid line represents the net effect of stage amplification effect and channel enlargement effect ($D_{50} = 0.30$ mm). The black lines from right to left (dash, solid, dot, dot-dash) represent $D_{50} = 0.25, 0.30, 0.35, 0.40$ mm, respectively. Correspondingly, the crossing-point discharges, where post-dam stages surpass the pre-dam values, are 7571, 6099, 5469, and 5164 $m^3\,s^{-1}$, respectively. Coarser beds lead to a smaller crossing-point discharge. Dot-dash vertical lines represent the flood discharges at different recurrence intervals (5, $10^2$, $10^3$, $10^4$ yr).

effects, with increasing water discharge, flood stage (post-damming) gradually catches up with, and ultimately surpasses, the pre-damming discharge stage (black line in Fig. 5c). It is because of this multiplier effect that, despite the magnitude of bed incision, resistance enhancement generated by larger dunes formed in a coarser-grained bed will generate a "crossing point", whereby post-damming flood discharge exceeds the flow stage of the equivalent discharge pre-damming condition. Continuous bed degradation will further enhance bed coarsening. Therefore, we perform a sensitivity analysis (Fig. 5c) to test the impact of a continuously coarsening bed. The crossing-point water discharge, whereby the post-dam stage surpasses the pre-dam stage, decreases consistently with the coarser bed grain size due to the intensification of the stage amplification effect, suggesting that

future bed coarsening will exacerbate the post-dam channel conveyance. Specifically, as $D_{50} = 0.25, 0.30, 0.35, 0.40$ mm, the corresponding crossing-point discharges are 7571 $m^3\,s^{-1}$ (95% CI [5696, 9550]), 6099 $m^3\,s^{-1}$ (95% CI [4847, 7889]), 5469 $m^3\,s^{-1}$ (95% CI [4408, 7022]) and 5164 $m^3\,s^{-1}$ (95% CI [4165, 6618]), respectively. Hence, as the bed of the LYR continues to coarsen, any flood discharge greater than the 5-yr recurrence interval (10,000 $m^3/s$) will generate a river stage that surpasses the pre-damming value, despite 3.44 m of bed degradation (as measured at Huayuankou) since completion of the XLD. We acknowledge that XLD water regulation causes flood discharges greater than the pre-dam bankfull value (4000 $m^3\,s^{-1}$) to be relatively rare. However, such events do occur, as was demonstrated recently (October, 2021). It is only a matter of time before the LYR

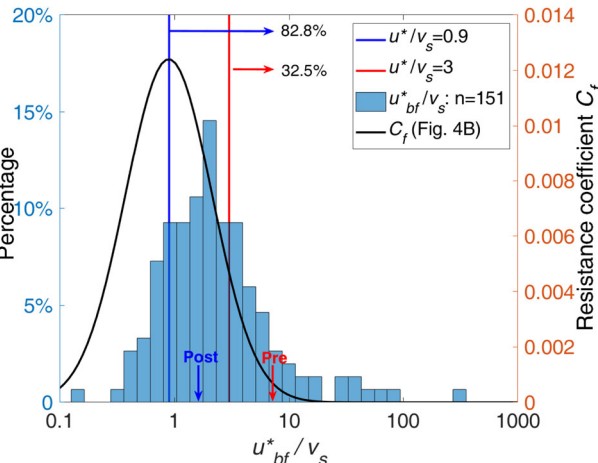

**Fig. 6 Distribution of bankfull suspension numbers for fine-grained rivers worldwide.** The vertical blue solid line represents the suspension number corresponding to the maximum resistance coefficient/dune size, obtained in the present study. This shows that 82.8% of fine-grained rivers maintain a bankfull suspension number greater than 0.9, indicating that for the majority of fine-grained rivers worldwide, bed coarsening will enhance dune size and thus flow resistance, so as to amplify flood stage. The vertical red solid line represents the suspension number (3.0) corresponding to borderline of silt-sand-bedded rivers (32.5% of fine-grained rivers worldwide), such as the Yellow River, where a bed coarsening process can act far more quickly over a short time scale, indicating a quick and strong flood amplification effect. Arrows on the x-axis indicate pre- and post-damming conditions of the lower Yellow River.

experiences sufficiently large floods, during which the downstream stage is unintentionally amplified, surpassing the pre-dam flood stage.

## Discussion

The model proposed herein for bedform height and suspension number, developed from extensive data collected from flume and field settings, is substantiated and extended by data from one of the world's largest dam-impacted fine-grained rivers, the lower Yellow River (LYR). Furthermore, the new model quantitatively outperforms other similar relations (Figs. 2, 4b, S2; Supplementary Tables S4, S5)[27,29,30]. The insight gained from this model is that for rivers where $u^*/v_s > 0.9$, bed sediment coarsening is expected to produce adjustments in dune size and increase flow resistance, which leads to an amplified flood stage. For $u^*/v_s$ values slightly greater than 0.9, the flood amplification effect will not be as obvious, or emerge as quickly, as compared to the LYR example. A global database of river bankfull hydraulic geometry (see Text "Global Bedform Database" in Methods)[54–56] is used to demonstrate that ~83% of fine-grained rivers are susceptible to amplification of flood stage (Fig. 6). Moreover, ~32.5% of fine-grained rivers are expected to behave similarly to the LYR, whereby bed coarsening due to damming could evolve rapidly due to the erosion of silt and very-fine sand[32,33] (Fig. 6). These results indicate the possible widespread existence of the phenomenon of degradation-induced flood-stage amplification.

Uncertainties remain surrounding field observations used to generate the predictive relation for bedform size, particularly for silt-sand-bedded rivers[28] (Fig. 2). For example, bedforms of silt- and sand-bedded rivers are irregular[32], so proper coverage in the longitudinal direction is necessary to capture accurate wavelength and height information. Additionally, uncertainties should be

properly documented, whereby data beyond simple mean values are reported, including for example, standard deviation (as used in Fig. 2) or, preferably, full distributions of dune height[28] (Supplementary Fig. S3). Existing relations between bedform size and hydraulic/transport parameters, including Froude number, suspension number and/or shear stress, are also subject to large uncertainties, partially due to the complexity of bedform geometry in natural rivers[28]. These factors also underline the challenge of reproducing silt-bedded bedforms in laboratory experiments[32]. A relation with less uncertainty would be facilitated by better documentation of bedform geometry (e.g., leeface angle)[28,39,57–60], incorporating physically the surface wave effect (Froude number) and water depth variation[40], and/or by assessing the impact of high sediment concentration[61,62] and bedform superimposition[59]. Our observations (Figs. 2, 4b), combined with previous studies[27,40], show that uncertainties can be constrained for a specific setting, and that the performance of the relation is improved by additional field data.

Previous studies have reported the possibility of both reductions and increases in flood stage due to fluvial morphological changes caused by human activities[63,64]. Studies showing an increase in flood stage have attributed this to the reduction in cross-sectional flow area due to loss of the channel-floodplain complex[8,63], as well as to the addition of spur dikes and other infrastructure that could contribute to channel roughness[64,65]. For the lower Yellow River, however, river training projects were implemented before construction of the XLD[66,67] and therefore do not explain the sudden change in bed material grain size (Fig. 1b, d), the development of bed roughness (Fig. 3), or increase in resistance (Fig. 4). Moreover, the implementation of river training projects such as spur dikes are relatively non-intrusive, with the aim to protect the bank and guide flow. In addition, the cross-sectional area of the channel increases due to bed erosion, so that neither loss of cross-sectional area nor the construction of river training projects necessarily contributes to the increase in flood stage. The analysis herein thus demonstrates that damming is the independent factor resulting in increased flood stage, because it drives bed coarsening, development of dunes, and enhanced overall flow resistance. It should be noted that dune fields can develop over the entire channel bed[49], so that they may exert more flow resistance than river training projects, indicating a potentially stronger control on flood-stage amplification. Amplification of flood stage due to damming has rarely been considered in flood-risk assessment, and thus our findings call for a careful evaluation of dam-impacted rivers where incremental increases in flood stage have been observed, but nevertheless could have been attributed to other factors[8,11,64,65].

It is difficult to overstate the importance of dams in terms of flood control. In many river basins, reservoirs are of critical importance for attenuating flood waves, and thus reducing flood peaks. Despite the fact that dams and reservoirs may reduce flood magnitude, the unanticipated loss of the flood-retaining capacity of the channel due to resistance enhancement can partially or completely offset efforts to reduce flood risk. Specifically, bed material coarsening and associated enhancement of flow resistance may worsen flood risk by amplifying flood stages when water impoundment in the reservoir is less effective during extreme floods. In the case of the lower Yellow River, if the resistance coefficient, $C_f$, increases by a factor of three (Fig. 4a), this necessitates that the Xiaolangdi Dam be operated so as to reduce its peak discharge by 57.7% ($1/\sqrt{3}$) to achieve a flood control similar to the pre-damming condition. The stark implication of this finding is that, during large floods, fine-grained channels downstream of dams should not be expected to operate

as effectively as the corresponding pre-dam channel in terms of routing flood water.

It is fortunate that the effect of flood-stage amplification has not yet created a major flood on the LYR. For the past two decades, the Yellow River catchment has been unusually dry[68] and the Xiaolangdi Reservoir has worked to effectively damp floods. However, climate models predict that precipitation in the Yellow River catchment will increase by 10 to 30% in the coming century[69]. Since our analysis shows an intensification of flood-stage amplification during large and extreme floods (Fig. 5b, c), as precipitation increases, such effects may emerge as a significant unintended consequence of damming. Meanwhile, the large amount of sediment input from the Loess Plateau has reduced capacity of the Xiaolangdi Reservoir to below 25% in just 20 years. For the coming decades, large floods combined with a lack of storage capacity could render the 88 million people living alongside the lower Yellow River at risk of severe flooding.

The effect of water and sediment from river tributaries on bed coarsening and flood amplification is not explicitly considered herein. However, the relevant analyses are straightforward within the proposed framework. The variation of bed grain size depends on the competition between fine-grained sediment supply from tributaries (or the reservoir itself) and bed coarsening caused by clear-water releases. With appropriate constraints on bed grain size and modeling efforts[18,19], the relation in Fig. 4b can be used to quantitatively determine potential flood amplification due to changes in bed grain size. One scenario that requires particular attention is bed coarsening combined with a large flood from downstream tributaries, whereby the reservoir has no ability to attenuate the flood wave. Such a scenario is not out of the realm of possibility: in the summer of 2020, the lower Yangtze River, China, experienced an unprecedented monsoon season[70], whereby a major flood originated from tributaries downstream of the Three Gorges Dam. A similar scenario occurred along the lower Yellow River in the summer of 2021, whereby an unforeseen flood formed from the tributaries downstream of the Xiaolangdi Dam due to extreme local precipitation (historically, the second largest). The contribution of tributary water input to flood-stage amplification, and associated risk factors, merits further studies.

We summarize our findings hereinafter. Coarsening of the bed material in the lower Yellow River has developed due to erosion of the channel bed downstream of the Xiaolangdi Dam (XLD). This coarsening coincides with a fundamental change in the stable state of bedforms on the riverbed, yielding dune configurations with heights and aspect ratios akin to other large, lowland sand-bedded rivers. This, in turn, increases bed roughness and raises the resistance coefficient by a factor of 3 (i.e., from 0.0015 to 0.0046), which amplifies the flood-water depth for a given water discharge (per-unit width) by a factor of 1.4 to 2.0. This is evidenced by field data collected downstream of the XLD: despite 3.44 m of channel bed degradation (and channel widening), the flood stage of the impacted channel bed at Huayuankou has decreased minimally for small (yearly) flood events. However, for moderate to large floods (i.e., >5-yr recurrence interval), flood stage is predicted to be greater than the pre-dam channel, despite substantial bed erosion, and thus flood risk is also higher.

A simple model shows that this phenomenon of dam-induced flood-stage amplification exists in ~80% of fine-grained rivers (bed grain size <2 mm) worldwide, with ~30% of fine-grained rivers susceptible to behavior similar to the lower Yellow River. Specifically, these rivers will experience significant flood-stage amplification due to rapid bed coarsening after damming. This heretofore-unforeseen phenomenon should be used to evaluate the effectiveness of dams as flood-control agents, and assess the risk to human populations, because habitation, infrastructure and

flood mitigation near many large, lowland rivers, depends on the ability of dams to attenuate, and not amplify, river floods.

## Methods

**S1. Hydraulic, cross-section and bed material database**. Since the 1950s, hydrologic surveys have been conducted routinely on the lower Yellow River (LYR), whereby substantial hydraulic data are collected, including water depth, surface slope, and flow velocity, as well as cross-sectional geometry and bed material grain size. At present, the hydrological stations operated by the Hydrological Bureau of the Yellow River Conservancy Commission collect water discharge and sediment concentration daily. Water surface slope is collected based on the stage differences at two referenced stage recorder stations on a relatively straight reach of the river, but the frequency of collections is lower than daily. Bed material samples are collected monthly, with a greater frequency during the flood season. Cross-sectional surveys are conducted at least twice per year, with one before and one after the flood season. Water velocity and sediment concentration measurements are collected at approximately 5 profiles or more per cross-section; for each profile, 3–5 velocity and water-sediment samples are taken to estimate the average velocity and sediment concentration. An acoustic Doppler current profiler (ADCP) is used for water discharge measurements when the sediment concentration is low; otherwise, a mechanical velocimeter is used for velocity measurements (which are subsequently converted to discharge). Multiple bed material samples are taken from the channel bed at the hydrologic station cross-sections; the reported grain size is the average value over the cross-section. Daily water discharge, average sediment concentration, water surface slope, cross-section geometry, and average bed grain size are published by the Hydrological Bureau of Ministry of Water Resources of China. The detailed standards can be found in ref. [71]; these generally comply with those used internationally[72].

We used hydraulic and cross-section information from the 1980s, because in the 1990s extensive water usage by communities along the LYR, for agricultural, industrial and municipal purposes, greatly reduced river runoff, triggering fine sediment deposition and associated bed aggradation. As a consequence, the data from the 1990s are not an accurate representation of natural pre-damming states, and are thus purposely excluded. We instead use the hydraulic and cross-sectional data from the 1980s to represent pre-dam conditions, so enabling a fair comparison.

**S2. Refined bed material surveys**. Since only cross-sectionally averaged grain size is published, and since the hydrologic stations are distant from each other (~100 km), bed material surveys based on a refined grid were conducted to determine the location where the Xiaolangdi Dam has impacted the bed. These surveys were conducted on floating bridges over the lower Yellow River, which are typically spaced 10–20 km apart. For each cross-section, 3–5 cores are collected with a tripod deployed from a survey boat or from the floating bridges. The core lengths range from 0.7 to 2.0 m, and samples along each core were taken every 0.1 m. For the purpose of our study, we only use the sediment samples collected from the bed surface of the channel thalweg, whose location is roughly determined by the line of greatest water surface flow velocity. The bed-surface sediment sample from the channel thalweg is usually the coarsest among those of all cores across cross-section. The median grain sizes of the surface sediment samples in the thalweg are presented in Fig. 1d. In a dynamic river such as the LYR, local grain size of the bed material may fluctuate due to changing contributions from eroded in-channel bars and banklines, as well as due to varying upstream/tributary sediment input. The grain-size variability from Xindian, via Changxing, to Dongming may be associated with these conditions, which would temporarily supply fine sediment to the thalweg at Xindian.

**S3. Equilibrium database on hydraulic and sediment transport**. The equilibrium database was selected from the extensive hydrologic data collected from the LYR using the follow criteria: (1) water discharge, sediment concentration, water surface slope, and the grain-size distribution of both suspended sediment load and bed material must be available at the same time (i.e., temporally consistent); (2) the water surface elevation, water discharge and sediment concentration must be stable for at least 2 h, so as to be deemed in equilibrium (i.e., steadiness); and (3) at similar discharges, sediment concentration and hydraulic factors must be comparable (i.e., repeatability). When these conditions were found to be satisfied, the pre-damming and post-damming databases were compiled and used to compare changes in hydraulic conditions[32,33].

**S4. Channel bathymetric surveys**. A Multibeam Bathymetric Echo Sounder (MBES)[49] and Parametric Echo Sounder (PES) were used for channel bed surveys on the LYR[73,74]. At Lijin, two MBES surveys were conducted during a flood flow in 2015, and base flow in 2016. At Huayuankou, the PES was used to measure the channel bed in 2016, and the MBES was used under base flow conditions in 2016 and 2017, and also under flood conditions in 2018. The best possible coverage of MBES (40 m ~ 100 m swath width, extending over multiple kilometers) was achieved, given the complexity of the flow and morphological conditions in the LYR. For example, since the LYR is only several meters deep along its thalweg and the MBES swath width is proportional to flow depth (c. 1–5 m in the LYR), the coverage of the MBES survey is limited. Therefore, it is neither efficient nor safe to

achieve complete coverage (for example, in order to produce the bed topography data presented herein, we suffered multiple strandings with damage to our boat and instruments). Besides, the need for additional coverage in portions of the river <1 m deep is unnecessary for our study, so we focus instead on the thalweg, which conveys the majority of water flow. The data cover a range of water depths and thus flow velocity conditions, which are key controlling parameters for bedform morphology. Therefore, the bedform coverage presented (i.e., width 40–150 m in the river greater than 1.0 m depth) serves to represent the typical reach-scale conditions of the LYR.

The vertical resolution of water depth is *c.* 1 cm where the MBES (millimetric resolution) was combined with RTK-resolved vertical position (millimetric to centimetric depending on the boat speed and satellite coverage). The horizontal resolution is better than 5 mm and for the most part near 1 mm. This vertical resolution is sufficient to capture the bedform height (typical size is 16–22 cm) at both Lijin and (70-90 cm) at Huayuankou. As to the horizontal coverage, two multibeam maps at Lijin shown in Figs. 3b, S3a have are *c.* 2.5 and 1.5 km along the river. The corresponding survey widths are 7 m and 42 m, respectively. The longitudinally stretched appearance of the bed profile in Figs. 3b, S3a is because low-relief bedforms have to be shown over a very long profile, whereas more typically sized dunes from Huayuankou can be showed over a relatively short distance. The relatively shallow water at Lijin constrained the channel area that could be surveyed.

**S5. Global Bedform Database.** The database utilized herein is a compilation from two studies, namely, Bradley and Venditti[39] and Naqshband et al.[40]. The former focused mostly on laboratory data, whilst the latter included field data. In the laboratory cases, water flow is steady and uniform (normal flow) and sediment transport is in equilibrium (no net erosion and deposition on the channel bed), so that bedforms developed naturally and fully covered the bed. Furthermore, bedform height, measured together with grain size and hydraulic information (including water depth, water surface slope, and mean flow velocity) were also required. Side-wall friction corrections were used to remove wall resistance, so that the only resistance was due to dunes covering the bed. For the field observations, bedform geometry was measured under the assumption that bedforms were in equilibrium with normal flow conditions. Compared with the flume experiments, hydraulic and channel geometry information from the field studies is less complete, with some records lacking mean flow velocity and/or flow surface width data.

**S6. Hydraulic prediction for the flow stage based on observed cross-sections.** In order to derive the relation between water discharge ($Q_w$) and water stage level ($Z$), the hydraulic relation between water depth ($H$) and flow velocity ($U$) (Eq. 1) needs to be stated and combined with information concerning the compound shape of the channel-floodplain cross-section. Since $Z = Z_b + H$ and $Z_b$ is determined by the observed cross-section, we can relate $Q$ with $H$ based on a variant of Eq. 1 designed for multi-threaded channel-floodplain complexes[25,26,75], expressed as:

$$Q_w = AU = A[gSA/(\Gamma C_f)]^{0.5} = g^{0.5}\left(\frac{A^3}{C_f\Gamma}\right)^{0.5}S^{0.5} = K\frac{g^{0.5}S^{0.5}}{C_f^{0.5}}, \quad (S1)$$

$$K = \left(\frac{A^3}{\Gamma}\right)^{0.5} = AR^{0.5}, \quad (S2)$$

$$K = \sum K_i = \sum\left(\frac{A_i^3}{\Gamma_i}\right)^{0.5}, \quad (S3)$$

where $A = \int W(h)dh$ is the flow area [L²], $W$ is the flow surface width [L] at depth $h$ (Supplementary Fig. S1a), $\Gamma$ is the wetted perimeter [L], $R = A/\Gamma$ is the hydraulic radius [L] and $K$ is the geometric conveyance factor [L^{2.5}]. In Eq. S1, the equation $\tau = \rho g SA/\Gamma = \rho u^{*2} = \rho C_f U^2$ is used; $C_f$ can be obtained from the relation given in Fig. 4b.

The conventional form of conveyance factor ($K_{ori} = K/C_f^{0.5}$) includes the resistance coefficient; however, we explicitly state the two terms in Eq. S1 such that we can attribute separately the effect of channel incision and widening and resistance change to the geometric conveyance factor $K$ and $C_f$ respectively.

For a compound channel such as illustrated in Supplementary Fig. S7, the channel-floodplain complex can be divided into multiple subchannels, and the geometric conveyance factor of channel subchannel $K_i$ can be summed into a total value for $K$ based on Eq. S3. Two examples with different flow stages $Z_1$ and $Z_2$ are shown in Supplementary Fig. S7, whereby their corresponding flow areas, channel widths, wetted perimeters are given and the conveyance factor $K_1$ and $K_2$ can be calculated accordingly.

As examples, we show detailed information of two calculations from 1981 to 2015, respectively (Supplementary Fig. S8). The same calculations were made for each year of the 20-year period (1981–1990 and 2006–2015). In Supplementary Fig. S8, the post-dam geometric conveyance factor, which is the major geometric control on flow conveyance (discharge), shows a significant increase, which results from a combined contribution from both channel incision and widening (Supplementary Fig. S8a–c). However, this increase is quickly offset and surpassed

by the effect of an increase in post-damming resistance caused by bed coarsening and enhancement of roughness (Supplementary Fig. S9). This indicates that the flood conveyance is only increased for small floods, but significantly reduced by the stage amplification effect in moderate and large floods.

## Data availability

The datasets generated and/or analyzed during the current study are provided in the article and the Supplementary Information file, and are also available from the corresponding authors upon reasonable request.

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

## Acknowledgements

We thank A. Moodie for post-processing the MBES data and Z.Li for his comments on extreme floods induced by climate change. We appreciate the careful and constructive reviews from Profs Samuel Munoz, Gordan Grant, and an anonymous reviewer. H.M., J.A.N., M.P.L., and G.P. gratefully acknowledge the NSF of the United States for support through Division of Earth Science (EAR) Grant 1427262. X.F. was supported by NSF of China through Grant 51525901. Y.Z. acknowledges support from NSF of China through Grant 51379087. Y.W.(a) acknowledges support from NSF of China through Grant 42041004. J.C. was supported by NSF Graduate Research Fellowship (grant no. DGE-1746047) and also the Department of Geology, University of Illinois, and the Jack and Richard C. Threet chair to J.B.

## Author contributions

H.M. designed the study, drafted the paper and was leading author of the paper. J.A.N., G.P., M.P.L., and J.B. provided substantial editorial feedback. J.A.N., H.M., and D.P. conducted the multibeam survey at Lijin. Y.Z., X.F., and Y.W.(a) conducted the multibeam survey at Huayuankou. J.B., J.C., and Y.Z. conducted the parametric echo-sounder bed surveys at Huayuankou. H.M., Y.W.(b), and B.W. analyzed the historical stage-discharge and cross-sectional data. J.C. processed the parametric echo sounding survey data and analyzed the bedform statistics. M.P.L. and H.M. developed the relations among dune height, resistance and suspension number. All authors participated in discussion and writing of the paper.

## Competing interests

The authors declare no competing interests.
