## [Peer Review File · Nature Communications]

Amplification of downstream flood stage due to damming of fine-grained riversReviewers' Comments:

Reviewer #1:

Remarks to the Author:

Review for 'Amplification of downstream flood stage due to damming of fine-grained rivers" by Ma et al., submitted to Nature Communications.

This study examines changes in channel morphology and roughness of the lower Yellow River to propose that on reaches downstream of a dam the channel bed becomes coarser, forms larger dunes, and thus amplifies river stages for larger flood events. The mechanism proposed here is interesting, novel, and supported by the evidence presented in the paper. I enjoyed reading the manuscript, and believe it presents an important step forward in developing a mechanistic understanding of how river infrastructure influences stages (and thus flood risk).

I have two minor suggestions to improve the study:

(1) It is clear that the bed has coarsened downstream of the dam, but could you please provide an explanation as to why this happens? This is partially addressed towards the end of the discussion, but it would be useful to briefly explain this to readers when the observation is first brought up (around line 113).

(2) There is a discussion (around line 242, and more in the paragraph starting on line 266) regarding the potential influence of dams on flood stages globally, and how this may be an underappreciated mechanism relative to levees/dykes that are more commonly invoked. My question here is: How far downstream from a dam do the authors believe their proposed mechanism is relevant? A brief (2-3 sentence) discussion of the scale of their proposed mechanism would be useful.

Otherwise, I believe this is a strong study and I support its publication in Nature Communications.

Samuel Munoz

Assistant Professor

Department of Marine & Environmental Sciences

Department of Civil & Environmental Engineering

Northeastern University

Reviewer #2:

Remarks to the Author:

Understanding flood behavior in response to dam regulation, particularly in the mega river, is of widespread importance due to the many valuable services it offers. Here Ma et al. explored the flood stage variation along the lower Yellow River after the construction of the Xiaolangdi dam through hydrologic and bathymetric surveys. Their results reveal that flood stage is lowered for small floods, but is amplified for moderate to large flood events, like >5-yr recurrence interval, because of rapid bed coarsening process after damming. The authors address a sound and important question; however, their conclusion is not supported by their surveys and they ignored the key differences between the small and large floods. I am afraid I can not recommend this work for publication until the following points are properly considered:

1. The authors indicate that sediment coarsening increases flow resistance and slows velocity relative to pre-dam condition, which have overruled the bed incision and amplified moderate and large floods while lowered small floods. However, flood at all levels will experience bed incision and sediment coarsening after dam construction, and therefore should have same responses. The main difference between small and large flood is the area where large flood covers, but small flood can not reach, rather than the river bed area. It is not reasonable to consider only the common area to get the

present conclusion. The non-common area that occupied by the large flood only should be the research focus as well, unfortunately, the authors ignored this in their work.

2. In Fig. 5B, it is obvious that post-dam flood water depth is lower than pre-dam for any given water discharge, including the flood events over 5-yr recurrence interval, which is inconsistent with the conclusion. How do you calculate the 5-yr recurrence interval? What is your database? It is better to show the floods corresponding to different time return periods in the figure.

3. How did you get the channel slope at Huayuankou? What is the difference between channel slope and water surface slope? The calculation of slope should have a reference point. Please justify.

Reviewer #3:

Remarks to the Author:

This provocative paper argues for a seemingly counter-intuitive finding: dams on fine-grained rivers can actually increase flood stages in downstream channels due to increased flow resistance as a consequence of bed coarsening. This conclusion challenges the widespread belief that dams overall decrease flooding, in part due to reduced flood heights downstream as a consequence of bed incision. If confirmed, the conclusions of this paper would be of great interest world-wide, as these results would force rethinking of the effects and consequences of dams at a time when there is a dramatic increase in dam construction in the developing world, and hydroelectric development is being promoted as a relatively "clean" energy source.

While the paper is generally well-written, logically laid out, and presents a reasonably compelling case, there are a number of critical issues with both the data and its interpretation that need to be resolved before acceptance and publication. Chief among them is the lack of discussion and consideration of the underlying physical mechanisms responsible for the phenomena observed. The authors document a coarsening of the bed of the Lower Yellow River (LYR) below Xiaolangdi Dam since the dam became operational in 1999, and link this to an empirical relation, derived from laboratory and field data, between suspension number and bedform height (Fig 2). Yet this is not a mechanistic linkage but only a proxy for one. The linkage is inferential and not well discussed. Presumably as the bed coarsens in the impacted reach, particle settling velocity (the denominator of the suspension number) increases, reducing the suspension number, which predicts greater bedform height, hence greater flow resistance.

I find this a rather convoluted and not entirely supported argument. Classical fluid dynamics theory shows that bedform type and scaling is largely a consequence of interactions between the free-surface and bed, as scaled by the Froude number. The suspension number (ratio of shear velocity (u^*) to settling velocity) has been shown to play a role (i.e., reference 37), but the flow regime, as defined by Fr , is a primary control. The authors acknowledge this, but I found their argument unconvincing. Moreover, since the x-axis of Fig. 2 is a ratio between u^* and suspension number, the interplay of both factors defines the hump-shape of the curve, yet the role of u^* in the data and relation is never explored. And u^* is highly dependent on discharge, which is not discussed, and it's not clear what u^* is being extracted from the database. As discussed in the line notes below, I calculated a 3-fold difference between the impacted and non-impacted reaches in u^* . So Fig. 2 is not just about bed coarsening.

The paper would be strengthened if the relation between bed coarsening and bedform shape were better developed. First, what is driving bed coarsening in the first place? While we know that stream beds below dams become armored due to reduced sediment supply, it's difficult for me to picture such fine-grained beds developing a true armor layer. Some discussion seems warranted. Second, what about bed coarsening drives increased bedform size? I can imagine an argument that might rest on the angle of repose of the larger grains, but there is no consideration of this in the paper.

The authors rightly choose to validate the prediction made by Fig. 2 by comparing bedform

morphology in both impacted (Huayuankou) and non-impacted (Lijin) reaches of the LYR. But the data comparison is problematic, due to the difference in resolution between the imagery and data from the two reaches (Fig. 3). Presumably this is due to differences between different multi-scan imaging employed (MBES vs. PES?). The impacted reach certainly shows beautiful large bedforms. Yet the coarseness of the imagery of the Lijin reach make the comparison unconvincing, a problem further exacerbated by confusing presentation in Fig. 3 (see notes). I certainly can appreciate the problems associated with getting this data (discussed in S4), but the authors need to confront the uncertainty of this measurement and discuss how it affects confidence in the comparison.

I found Fig 5 to be quite compelling, with one big caveat. The authors report that they did not see a relation between the resistance coefficient C_f and specific discharge, yet Fig 5 is all about the presumed linkage between Q and H via changing flow resistance. If C_f doesn't vary with q , then how do you explain this? More broadly, if bedform shape and size were the dominant controls on flow resistance, I would expect a strong dependency on q , since as flows increase, the relative submergence increases as well, reducing the potential impact on flow resistance. This needs to be addressed.

More comments below, keyed to line number. In spite of the issues I've raised, which deserve attention, I think the authors are on to something here, and the article deserves serious consideration after the problems are addressed. It has the potential to have a major impact on the field of geomorphic assessment of the effects of dams on rivers.

The authors are free to contact me if they have any questions about my review
Gordon Grant

49: "Engineered rivers" the same as dammed rivers?

53-54: Why "increasingly difficult"?

137: Define "granular bed". By definition, a bed made up of grains of material should be "granular".

136-139: Reword for clarity; it's a bit of a run-on sentence.

147: Here and in Fig. 2 and S5 there is no mention of the grain-size of sediment used to develop this parabolic relationship. What is the range of grain sizes used to develop this empirical curve and how does this compare with the LYR bed?

149-151: Following up on 147 comment above, I am puzzled by how the relationship between grain size and bedform height is being treated here. I understand that you have shown a bed coarsening in the impacted reach, and Fig. 2 predicts a difference in bedform height as a function of the changing suspension number, which is a dimensionless ratio between shear and settling velocity (u^*/v_s). I think you're arguing that for a constant u^* , an increase in grain size will result in a decrease in suspension number...I suspect that's what the arrow labelled "bed coarsening" is intended to show in Fig. 2. But is u^* constant for the impacted and non-impacted reaches?

From back-of-envelope considerations and data presented, this doesn't seem to be the case. You report (Lines 151-2) u^*/v_s for Huayuankou as $\sim 1-2$ while ~ 10 at Lijin. The corresponding mean grain size for these two sites is 0.21 and 0.10 mm respectively (Fig. 1D). Empirical relationships between grain size and settling velocity show a relation $v_s = aDb$ with $b \sim 1.5$ (e.g., Sternberg and others, 1999). From these numbers, we would calculate u^* at Huayuankou to be roughly 3 times that at Lijin. If this is correct, how is this factored into the analysis.

Finally, it's not clear to me what value of u^* you are using, since it changes with discharge. You may

have stated this already, but this dependency of $u^* = f(Q)$ needs to be explored.

154-156: I find this sentence a little confusing. As I understand it, you are primarily using the suspension number as a measure of flow intensity, not because it describes the underlying mechanism resulting in formation of dunes or plane beds. Thus "attributing" the formation of dunes to the suspension number seems off. The Froude number is, in my view, more closely tied to the mechanism of dune formation, since it describes free-surface effects, and is traditionally the basis on which flow and bedform are discriminated (e.g., Simon and Richardson, etc.). But of course, the Froude number refers only to flow dynamics and doesn't include sediment size at all, the latter being central to your argument.

157: I consider an Fr of 0.7-0.8 to be relatively high. We've seen standing waves beginning to form in sand-bed channels with Fr in this range. In fact, the paper you cite by Naqshband and others (2014; Ref. 37) defines a range of "low" Froude numbers as 0.05-0.32 while "high" Froude numbers range 0.32-0.84.

159: I agree with the point of this sentence and would argue that just showing where the two sites plot on Fig. 2 only suggests or predicts change in dune height; it does not provide evidence of "substantial change". The latter needs direct observation to confirm, as you suggest.

162-167: I found the data presented in Fig. 3 problematic, and less convincing than I had hoped. To begin with, the scales of the two reaches are completely different, and it is hard to determine whether the absence of clear dunes at Lijin is real or a consequence of the resolution of the imagery/measurement. Could you even detect bedforms if they were present at Lijin using the imagery you show? At a minimum, a clear discussion of the likely error associated with the coarser resolution seems warranted. I had great difficulty orienting where the data was actually taken from in each reach using the insets and profile lines. Do you interpret the convex up long profile at Lijin as reflecting a long wavelength, low amplitude form (your reference to the L/H ratio suggests this), yet the B-B' inset shows a wide range of wavelengths and amplitudes? You might have seen a closer resemblance between the two reaches had you chosen a different section to highlight. This figure shows the crux of your argument and needs to be much more compelling than it is.

196-197: See note for lines 638-9...I don't think your data actually extends this relationship

202-203: It is somewhat surprising to me that there is no relation between C_f and discharge (lines 191-192). If form drag due to bedforms is a primary contributor to overall resistance, one would expect that this resistance term would vary as a function of relative submergence which is directly proportional to discharge. Why doesn't it, either for the pre- or post-dam case when presumably, the shape of the bedforms is changing?

205-206: Remove passive voice

231-243: I find Figure 5 compelling, and yet it raises question. Considering that you find no relationship between C_f and specific discharge in the data, as discussed above, how can you then argue for a discharge dependency here? Unless I missed something, this strikes me as implausible, or at least requiring explanation. Moreover, you argue (239-240) for a coupling between bed incision and bed coarsening. I don't see any proposed mechanism to explain this, which gets at a larger point: the lack of a physical mechanism linking coarsening and resistance through bedform development. I think you've made a reasonable case for a correlation between the two, but there's been no discussion (up to now) as to how? This question of how applies to both bed coarsening and its link with bedform development. Is coarsening the consequence of armor development (which seems quite difficult to visualize in such fine-grained rivers)? Or is depletion of the fine fraction or...? And how does this lead to larger or steeper bedforms? These questions need to be introduced and explored.

258-265: I find this entire section confusing. First, what do you mean by "transport stage parameter" (line 259)? I'm also not clear what problem this enumerated list is targeted at, i.e., what exactly are all these refinements intended to help illuminate?

299: I suggest using the word "routing" instead of "retaining" flood water.

360-362: Here or in S2, briefly describe methods of sampling bed material, i.e., how was coring done?

381-384: Rephrase for clarity. Also, is stratigraphy measured in the cores, or did you just assume that the coarsest fraction represented the surface layer? How deep is the surface layer?

421: What do you mean by "normal flow and sediment transport equilibrium"?

638-39: Logic unclear; since you are using independently measured bedform height as a way of testing whether the predicted empirical relation is borne out, you can't really say that it allows you to extend the data field for that empirical relation as well.

References cited

Sternberg, R. W., Berhane, I., & Ogston, A. S. (1999). Measurement of size and settling velocity of suspended aggregates on the northern California continental shelf. *Marine geology*, 154(1-4), 43-53.

Responses to the comments and reviews on “Amplification of downstream flood stage due to damming of fine-grained rivers” by Ma et al.

We deeply appreciate the reviews from the three reviewers, hereby referred to as R1, R2, and R3, and have sought to revise the manuscript in accordance with their very helpful suggestions.

Specifically: (1) we have added discussion regarding the mechanism that drives bed coarsening after damming and its time-dependent nature. (2) We have implemented a mechanistically-based model to predict theoretically the relation between flow discharge and stage based on newly compiled 20-yr cross-sectional data. The mechanistically-based model has no adjustable parameter and does not use any data for calibration, and is therefore a purely predictive model. Although the mechanistically-based model is implemented to replace the previously-used empirical model, our conclusion that the post-dam flow stages of moderate to large floods surpass pre-dam stages remains unchanged, further demonstrating the soundness of our work. (3) We have added discussion regarding the reasoning for the hump-shaped relation between suspension number and bedform size, i.e., a mechanism of dune wash-out. (4) We have added discussion regarding how the uncertainties of field measurement and predictive modeling results could occur. We believe these changes have improved the manuscript, and are pleased that these new analyses have confirmed our previous conclusions and implications. Herein, we provide a detailed point-by-point response to the reviewers' comments.

REVIEWER COMMENTS

Reviewer #1 (Remarks to the Author):

Review for 'Amplification of downstream flood stage due to damming of fine-grained rivers' by Ma et al., submitted to Nature Communications.

This study examines changes in channel morphology and roughness of the lower Yellow River to propose that on reaches downstream of a dam the channel bed becomes coarser, forms larger dunes, and thus amplifies river stages for larger flood events. The mechanism proposed here is interesting, novel, and supported by the evidence presented in the paper. I enjoyed reading the manuscript, and believe it presents an important step forward in developing a mechanistic understanding of how river infrastructure influences stages (and thus flood risk).

→We very much appreciate these positive comments.

I have two minor suggestions to improve the study:

(1) It is clear that the bed has coarsened downstream of the dam, but could you please provide an explanation as to why this happens? This is partially addressed towards the end of the discussion, but it would be useful to briefly explain this to readers when the observation is first brought up (around line 113).

→We thank R1 for this reminder. The mechanism of bed coarsening after damming is selective entrainment of sediment grains. Different size groups of bed material sediment have different responses to the same shear stress exerted by the water flow, and as the upstream sediment supply is cut off, finer material is preferentially transported, thereby leaving coarser sediment and causing the median size of bed material to increase (i.e., bed coarsening). Details regarding selective transport of silt-sand sediment can be found in recent work by Naito et al. (2019) and An et al. (2021). We have now highlighted this mechanism in the second paragraph of the main text.

(2) There is a discussion (around line 242, and more in the paragraph starting on line 266) regarding the potential influence of dams on flood stages globally, and how this may be an underappreciated mechanism relative to levees/dykes that are more commonly invoked. My question here is: How far downstream from a dam do the authors believe their proposed mechanism is relevant? A brief (2-3 sentence) discussion of the scale of their proposed mechanism would be useful.

→ These are insightful questions and good suggestions for helping us modify the manuscript. As shown in several recent papers (e.g., Naito et al. 2019; An et al. 2021) and documented in a field survey report (Williams & Wolman, 1984), the bed coarsening effect after damming exists over a spatial extent that is time-dependent. When sediment input is cut off, the bed-coarsening wave is initialized, and it propagates downstream with time. Typical time scales for the lower Yellow River are assessed by noting the change in median grain size after closure of the dam. For example, at Huayuankou (200 km downstream of the XLD dam), it took 3-5 years for coarsening to be expressed (as we discussed in Ma et al., 2020). This was numerically verified in a recent model (Naito et al. 2019). Other systems also offer insight: for the Mississippi River and its tributary, it took less than three years for bed coarsening to appear 85 km downstream of Gavins Point Dam (Williams & Wolman, 1984); for the same river, a recent modeling study (An et al., 2021) showed that it would take 30-40 yrs for the onset of bed coarsening (initiated by Gavins Point Dam) to reach Memphis (RK1182). Similar results for bed coarsening were found for the Colorado River after several dam closures, typically taking less than 5 yrs to travel 20 km downstream (Williams & Wolman, 1984). In summary, when initiated, the bed coarsening wave propagates downstream at a rate of 10-70 km/yr, depending on the boundary and morphodynamic conditions.

As to why dam impact is “an underappreciated mechanism relative to levees/dykes that are more commonly invoked for stage amplification in large rivers”, we suggest that it has been difficult to link directly the cause for stage amplification with a dam impact because there is a time lag between dam completion and stage amplification. Furthermore, robust bedform geometry measurement techniques were far from mature before the 2000s, when multibeam echo sounder (MBES) systems were coming into mainstream use. It is difficult to conduct a comparative study on riverine bedforms for pre- and post-dam conditions in many rivers, particularly since the largest boom in dam construction globally was completed between the 1960s-1970s (please see fig. R1 below). However, renewed megadam construction in the last decade suggests the need for these effects to be widely considered.

The field site choice of the Lower Yellow River is thus critical for this research because it is currently undergoing a transition from pre- to post-dam conditions, with a rapid pace and striking geomorphic changes. Combining the right field conditions with advanced surveying techniques provides the novel opportunity to demonstrate the phenomena of bed-coarsening, bedform enhancement, and therefore flood stage amplification due to damming.

Inspired by these questions raised by the referee, we have added some of the above discussion to the manuscript.

Figure R1. Global dam construction over the past 100 years, Source: Global Reservoir and Dam (GRanD) Database. (From Yang & Kelly, 2015)

Otherwise, I believe this is a strong study and I support its publication in Nature Communications.

→We are deeply grateful for your recommendation.

Samuel Munoz

Assistant Professor

Department of Marine & Environmental Sciences

Department of Civil & Environmental Engineering

Northeastern University

Reviewer #2 (Remarks to the Author):

Understanding flood behavior in response to dam regulation, particularly in the mega river, is of widespread importance due to the many valuable services it offers. Here Ma et al. explored the flood stage variation along the lower Yellow River after the construction of the Xiaolangdi dam through hydrologic and bathymetric surveys. Their results reveal that flood stage is lowered for small floods, but is amplified for moderate to large flood events, like >5-yr recurrence interval, because of rapid bed coarsening process after damming. The authors address a sound and important question; however, their conclusion is not supported by their surveys and they ignored the key differences between the small and large floods. I am afraid I can not recommend this work for publication until the following points are properly considered:

→We thank the reviewer for recognizing the soundness and importance of the proposed question. We have sought to clarify our manuscript in light of their concerns.

1. The authors indicate that sediment coarsening increases flow resistance and slows velocity relative to pre-dam condition, which have overruled the bed incision and amplified moderate and large floods while lowered small floods. However, flood at all levels will experience bed incision and sediment coarsening after dam construction, and therefore should have same responses. The main difference between small and large flood is the area where large flood covers, but small flood can not reach, rather than the river bed area. It is not reasonable to consider only the common area to get the present conclusion. The non-common area that occupied by the large flood only should be the research focus as well, unfortunately, the authors ignored this in their work.

→ We thank the reviewer for raising this concern. We wish to clarify that in our analyses, we did not invoke the assumption that both small and large floods occur only within a common area, as described by R2. Our analyses, regarding flow depth and flow stage, consist of two parts as shown in Figs. 4c and 5b. For the first part, we consider a mechanistic analysis whereby the relation between water discharge per-unit-width and water depth (i.e., q_w - H relation) is based on a theoretical hydraulic relation between the two variables (i.e., the hydraulic resistance relation, where both q_w and H are averaged over channel width). Even in this theoretical analysis, we did not invoke the assumption that “both small and large flood present only on the common area”. Instead, the small flood is averaged over a narrower width and the large flood over a wider width, as much as they actually occupy the channel. The theoretical and data analyses together show that the width-averaged behaviors of the q_w - H relation indeed have distinctly different pre-dam and post-dam conditions, as we predict based on the bed roughness change. This evidence is crucial because it provides both a theoretical foundation and direct evidence that the flow resistance changed significantly from pre-dam to post-dam.

We then applied this verified theoretical width-averaged assessment to include bed degradation and changes to cross-sectional shape. In the previous version of the manuscript, we empirically developed a Q_w - Z relation (total water discharge Q_w and stream stage Z) based on data, whereby Q_w and Z incorporated channel width variation as well as bed degradation. The power-law Q_w - Z relation, although empirical, is a common practice and has hydraulic basis (i.e., Petersen-Øverleir, 2006). Nevertheless, we recognize that the empirical means to develop the Q_w - Z relation is less convincing than a theoretical derivation.

Thus to further test the relations between Q_w - Z for pre- and post-damming conditions, we now develop a theoretically-derived hydraulic model for compound channels based on 20 years of field data from the LYR. Since no observed Q_w - Z data were used to calibrate this new relation, the modeling results are theoretical and predictive, and can therefore be directly compared to, and verified by, observed data (Fig. R2). Please note that the theoretically-derived relation shows no significant difference from the previous empirical relation. Consistent with our previous conclusion, the post-dam flood stage will surpass the pre-dam stage at small floods, in particular, with a discharge more than the 5-yr recurrence flood (10,000 m^3/s).

Figure R2. Comparison between theoretical predictions and observations. Theoretical predictions are based on the 20-yr cross-sectional data and the mean bed grain size as input. The Qw-Z data are not used to calibrate any parameters. The model shows that at $c. 6000 \text{ m}^3/\text{s}$ the post-dam stage surpasses the pre-dam stage. With potential uncertainties of cross-section variation and grain size, this value may vary: we have thoroughly discussed this in the revised manuscript. The range for this value is always less than 5 yr recurrence interval flood ($10000 \text{ m}^3/\text{s}$).

We have added an example to illustrate the consideration of channel incision and widening, and for small and large floods, in the pre- and post-dam channel, as shown in fig. R3. The detailed modeling information can be found in the modified *Methods*. The observed cross-sectional data were used to show the relations of channel width, wetted area, wetted perimeter, hydraulic radius, and geometric conveyance factors, with respect to different flow stages. Fig. R3A demonstrates channel incision and widening. All hydraulic-related geometric factors are calculated based on these observations. The relevant dynamic factors are shown below in Fig. R9

Figure R3. Examples of computations of channel geometry and geometric conveyance factor with respect to flow stage. The pre-dam case is computed from the cross-sectional data of 1981, and post-dam case is computed from data of 2015 (Fig. 5A). Post-dam channel widening and incision can be observed in Panel A and as a result, the post-

dam wetted area (B), perimeter (C) and geometric conveyance factor (D) all increase significantly. (Fig. S8 in the revised manuscript)

We have added these results into the revised manuscript. The detailed procedures to combine the theoretical q_w - H relation with cross-sectional data to obtain the Q_w - Z relation have been provided in the Method and Supplementary Information.

2. In Fig. 5B, it is obvious that post-dam flood water depth is lower than pre-dam for any given water discharge, including the flood events over 5-yr recurrence interval, which is inconsistent with the conclusion. How do you calculate the 5-yr recurrence interval? What is your database? It is better to show the floods corresponding to different time return periods in the figure.

→ We have clarified that the water stage Z in Fig. 5B is not the water depth but actually the sum of water depth h and bed elevation z_b ($Z=h+z_b$). The water depth analysis is shown in Fig. 4. We have theoretically derived the new Q_w - Z relation, verified by field data, which again supports our statement that the moderate and large floods, post-damming, will produce greater water levels than pre-damming. We have provided results for floods with different recurrence intervals (5, 100, 10^3 and 10^4 yr) in Fig. 5C, whereby the recurrence interval is based on the extensive hydrologic evaluations of robust historical data from the Yellow River Conservancy Commission (YRCC, 2013).

3. How did you get the channel slope at Huayuankou? What is the difference between channel slope and water surface slope? The calculation of slope should have a reference point. Please justify.

→ Thank you for the question. We have clarified the procedure to obtain the water surface slope in *Methods*. The water surface slope was measured and used in the study. Two reference points were set at a relatively straight reach of the river (surveyed region), and two surveyed stream stage markers were placed at each point to continuously record stage level within a measuring period. In the data of 1980s, two physical markers were used, and reading of the number was conducted by surveyors in the same measuring period. These markers were subsequently replaced by ultrasonic gauges. The water surface slope is calculated based on the difference in water level at the same time at each station divided by the streamwise distance between them.

Reviewer #3 (Remarks to the Author):

This provocative paper argues for a seemingly counter-intuitive finding: dams on fine-grained rivers can actually increase flood stages in downstream channels due to increased flow resistance as a consequence of bed coarsening. This conclusion challenges the widespread belief that dams overall decrease flooding, in part due to reduced flood heights downstream as a consequence of bed incision. If confirmed, the conclusions of this paper would be of great interest world-wide, as these results would force rethinking of the effects and consequences of dams at a time when there is a dramatic increase in dam construction in the developing world, and hydroelectric development is being promoted as a relatively “clean” energy source.

→ We deeply appreciate the reviewer’s positive evaluation.

While the paper is generally well-written, logically laid out, and presents a reasonably compelling case, there are a number of critical issues with both the data and its interpretation that need to be resolved before acceptance and publication. Chief among them is the lack of discussion and consideration of the underlying physical mechanisms responsible for the phenomena observed. The authors document a coarsening of the bed of the Lower Yellow River (LYR) below Xiaolangdi Dam since the dam became operational in 1999, and link this to an empirical relation, derived from laboratory and field data, between suspension number and bedform height (Fig 2). Yet this is not a mechanistic linkage but only a proxy for

one. The linkage is inferential and not well discussed. Presumably as the bed coarsens in the impacted reach, particle settling velocity (the denominator of the suspension number) increases, reducing the suspension number, which predicts greater bedform height, hence greater flow resistance.

→ We thank the reviewer for raising these concerns. We have responded to them one-by-one below. We have clarified the underlying physical mechanisms responsible for the observed phenomena and have incorporated these clarifications into the revised manuscript.

I find this a rather convoluted and not entirely supported argument. Classical fluid dynamics theory shows that bedform type and scaling is largely a consequence of interactions between the free-surface and bed, as scaled by the Froude number. The suspension number (ratio of shear velocity (u^*) to settling velocity) has been shown to play a role (i.e., reference 37), but the flow regime, as defined by Fr, is a primary control. The authors acknowledge this, but I found their argument unconvincing.

→ We appreciate R3's comment on this issue. A detailed response can be found below.

Moreover, since the x-axis of Fig. 2 is a ratio between u^* and suspension number, the interplay of both factors defines the hump-shape of the curve, yet the role of u^* in the data and relation is never explored. And u^* is highly dependent on discharge, which is not discussed, and it's not clear what u^* is being extracted from the database. As discussed in the line notes below, I calculated a 3-fold difference between the impacted and non-impacted reaches in u^* . So Fig. 2 is not just about bed coarsening.

→ We appreciate R3's comment on this issue. Detailed responses can be found below in the one-by-one responses.

The paper would be strengthened if the relation between bed coarsening and bedform shape were better developed. First, what is driving bed coarsening in the first place? While we know that stream beds below dams become armored due to reduced sediment supply, it's difficult for me to picture such fine-grained beds developing a true armor layer. Some discussion seems warranted. Second, what about bed coarsening drives increased bedform size? I can imagine an argument that might rest on the angle of repose of the larger grains, but there is no consideration of this in the paper.

→ We thank R3 for summarizing above concerns into two questions. We will answer these questions here; to be sure, we have added text to the revised manuscript to help clarify these concerns.

For bed coarsening, sediment-depleted water (i.e., "clear water") released from dams preferentially transports finer sediment. This phenomenon, termed selective entrainment, is well-documented, and produces an increase in median bed grain size increment (i.e., bed coarsening), as has been illustrated in recent work by Naito et al. (2019) and An et al. (2021).

As to the dune height correlation with suspension number, we agree with R3 that the relevant fluid dynamics theory on this point is still maturing, although both the correlation between dune height and suspension number, and the use of a suspension number to demarcate bedform state, have been empirically used and extensively verified (e.g. van Den Berg & van Gelder, 1993).

Indeed, the Froude number (Fr) provides an important control on bedform states; however, large Fr numbers usually occur in steep channels and are relatively rare in lowland, fine-grained systems, which are the main focus of this study. For instance, Fr in most of records from the LYR is below 0.5, which is far less than the transition threshold (0.8) required to generate upper regime plane bed. Interestingly, the LYR is prone to relatively higher Fr compared to other large, lowland rivers because the slope of LYR is up to 8×10^{-4} (i.e., relatively high). Bradley and Venditti (2019) illustrated that dune height $H(m)$ and steepness (i.e., H/L , ratio of dune height to wavelength) are not found to be significantly decreased until $Fr > 0.6$, which rarely occurs (if ever) in lowland, fine-grained rivers (Figure R4 top panels). This study also showed that at suspension number ($Sus = u^*/v_s$, ratio of shear velocity to settling velocity of sediment grain) > 1 , bedform size and steepness drop rapidly (Figure R4 bottom panels) and $Sus > 1$ is a condition

that most fine-grained rivers (~80% as shown in Fig. 6 of the main text) reach even under small floods (bankfull). This indicates that for lowland, fine-grained rivers, the suspension number is the predominant factor determining the bedform size.

Figure R4. Dimensional dune height (H in meters) and steepness (H/L , L is dune wavelength) versus Froude number (Fr ; top), and suspension number (bottom). These are Figures 12 and 13 of Bradley & Venditti (2019).

Recent progress in fluid dynamics theory, as well as observations from the lab and field, have provided additional evidence that the suspension number is directly related to bedform size, and is likely the predominant control of bedform morphology in the lowland fine-grained rivers, where variability of the Fr number is relatively small.

From fluid dynamics theory, Andreotti et al. (2011) derived that, in either end of Fr (i.e., $\rightarrow 0$ and $\rightarrow \infty$), the saturation length of sediment transport, $L_{sat} \propto (HU)/v_s = Sus * H / (C_f)^{0.5}$ (where H is water depth, U is flow velocity, v_s is the sediment settling velocity, and Sus is the suspension number), provides a critical control on the wavelength of bedforms. Specifically, 2-D ripple/dune increasingly tilts in a transverse direction as L_{sat}/H increases (i.e., towards chevron/linear dunes / alternate bars, which have much longer bedform lengths – and thus relatively low-relief – compared to standard sized dunes/ripples). This is best illustrated in Fig. R5, from Andreotti et al. (2011), whereby the increment in $L_{sat}/H (\sim Sus / (C_f)^{0.5})$ leads to a longer wavelength:

Figure R5. Wavenumber k_m ($k_m=2\pi/L_b$ and L_b is the longitude wavelength of the bed form) and the most unstable mode as a function of L_{sat}/H , in the limit $Fr \rightarrow 0$. The solid line corresponds to $u^*/u^*_c \rightarrow \infty$ and the dotted line to the limit $u^*/u^*_c \rightarrow 1$. (Fig. 7b of Andreotti et al. 2011)

Another well-accepted hypothesis is that as the suspended sediment concentration increases (i.e., large Suspension number), turbulence modulation, including stratification effect and interactions between vicious sublayer and fine particles, may significantly damp turbulent shear stress (Reynolds stress) near the bed, thereby suppressing the emergence of large and/or steep dunes (e.g., Best, 2005). This hypothesis, although not yet derived from fluid dynamics theory, was recently empirically verified, whereby studies included:

(1) lab experiments (Nashqband & Hoitink, 2020) with Fr numbers well below the conventional threshold to upper regime plane bed (i.e., $Fr < 0.8$), but still showing a hump-shape relation between dune height and the suspension number, and exhibiting a dominance of suspension number on the dune size even for small Fr (Fig. R6 from Nashqband & Hoitink, 2020);

Figure R6. Relation between the relative roughness (y-axis; ratio of dune height to water depth) and suspension number (x-axis). The yellow squares represent shallow-flow laboratory experiments with lightweight polystyrene particles, whereby $0.17 < Fr < 0.30$, i.e., well below the transition threshold, but the data show clearly a hump shape with a transition to upper regime plane bed. The red triangles represent laboratory experiments with $0.32 < Fr < 0.84$, and the blue circles show dune height data from seven deep river data sets ($0.10 < Fr < 0.30$), all documented in Naqshband et al. (2014) and used in this study. Both show a clear hump shape relation until upper-regime plane bed is reached, particularly in field cases. (Fig. 3 in Nashqband & Hoitink, 2020)

(2) experiments that demonstrate an increased suspended load enables sediment to bypass the dune crest, and instead of being deposited to dune trough so as to migrate the dune stably, instead directly contributing to dune field wash out (Nashqband et al., 2017); and,

(3) field studies showing that high suspended sediment flux can lead to a “flattening” of bedforms (Best 2005).

With the above evidence, we thus postulate that the suspension control on the bedform shape is demonstrated clearly. Moreover, we have reflected this in the revised manuscript.

The authors rightly choose to validate the prediction made by Fig. 2 by comparing bedform morphology in both impacted (Huayankou) and non-impacted (Lijin) reaches of the LYR. But the data comparison is problematic, due to the difference in resolution between the imagery and data from the two reaches (Fig. 3). Presumably this is due to differences between different multi-scan imaging employed (MBES vs. PES?).
→We will address this concern below.

The impacted reach certainly shows beautiful large bedforms. Yet the coarseness of the imagery of the Lijin reach make the comparison unconvincing, a problem further exacerbated by confusing presentation in Fig. 3 (see notes). I certainly can appreciate the problems associated with getting this data (discussed in S4), but the authors need to confront the uncertainty of this measurement and discuss how it affects confidence in the comparison.

→We appreciate this very good suggestion, and have added text to the Discussion regarding the uncertainty around acquiring bedform statistics. The coarseness of imagery is a result of the elongated profile (with a relatively narrow coverage) of the highly irregular shaped bedform at Lijin, not a problem of resolution, which is discussed further in responses below.

I found Fig 5 to be quite compelling, with one big caveat. The authors report that they did not see a relation between the resistance coefficient C_f and specific discharge, yet Fig 5 is all about the presumed linkage between Q and H via changing flow resistance. If C_f doesn't vary with q , then how do you explain this? More broadly, if bedform shape and size were the dominant controls on flow resistance, I would expect a strong dependency on q , since as flows increase, the relative submergence increases as well, reducing the potential impact on flow resistance. This needs to be addressed.

→We apologize this was not stated clearly. According to the relation between H_d and u^*/v_s , C_f should vary with q , which agrees with R3's assessment. However, in the original fig. 4A, because of the sparsity of the data, the trend, although emerging as expected, was not statistically significant when plotted as pre-dam and post-dam data, respectively. But the trend is quite clear as all data are plotted together in Fig 4b. We thus have changed the representation of the data in the modified Fig.4a. We thank R3's for their reminder on this ambiguous statement. As to the reason for flow resistance change, our analyses indicate that the bed grain size change between pre- and post-dam conditions plays the most important role for setting resistance, while the shear velocity plays as a secondary role. This assessment is comprehensively discussed below.

More comments below, keyed to line number. In spite of the issues I've raised, which deserve attention, I think the authors are on to something here, and the article deserves serious consideration after the problems are addressed. It has the potential to have a major impact on the field of geomorphic assessment of the effects of dams on rivers.

→We appreciate the assessment and have made significant revisions based on the input.

The authors are free to contact me if they have any questions about my review
Gordon Grant

49: “Engineered rivers” the same as dammed rivers?

→We used “Engineered rivers” as a general term referring to rivers with one or many river-engineering structures, such as levees, dikes, dams, etc. We have clarified this in the revised manuscript.

53-54: Why “increasingly difficult”?

→We have deleted this.

137: Define “granular bed”. By definition, a bed made up of grains of material should be “granular”.

→Done

136-139: Reword for clarity; it’s a bit of a run-on sentence.

→Done

147: Here and in Fig. 2 and S5 there is no mention of the grain-size of sediment used to develop this parabolic relationship. What is the range of grain sizes used to develop this empirical curve and how does this compare with the LYR bed?

→Good question. The grain size range of the global database is from 0.13 mm to 36 mm, with a focus on sand-bedded cases (80%). It thus overlaps the grain size range of the pre-dam LYR bed, whose grain size is generally between 0.05 mm and 0.15 mm, and the post-dam grain size range (from 0.15 to 0.35 mm), as well as the grain size range of the global database. Although the grain size range of the pre-dam LYR bed is not entirely within the data range of the global bedform database, our observations indicate that they agree with the empirical relation, and also fill in the previously-unexplored territory of the existing data and relation. Thanks to the Reviewer’s question, we have added a description of grain size range in the revised manuscript and Supplementary Information (Table S2).

149-151: Following up on 147 comment above, I am puzzled by how the relationship between grain size and bedform height is being treated here. I understand that you have shown a bed coarsening in the impacted reach, and Fig. 2 predicts a difference in bedform height as a function of the changing suspension number, which is a dimensionless ratio between shear and settling velocity (u^*/v_s). I think you’re arguing that for a constant u^* , an increase in grain size will result in a decrease in suspension number...I suspect that’s what the arrow labelled “bed coarsening” is intended to show in Fig. 2. But is u^* constant for the impacted and non-impacted reaches?

From back-of-envelope considerations and data presented, this doesn’t seem to be the case. You report (Lines 151-2) u^*/v_s for Huayuankou as $\sim 1-2$ while ~ 10 at Lijin. The corresponding mean grain size for these two sites is 0.21 and 0.10 mm respectively (Fig. 1D). Empirical relationships between grain size and settling velocity show a relation $v_s = aDb$ with $b \sim 1.5$ (e.g., Sternberg and others, 1999). From these numbers, we would calculate u^* at Huayuankou to be roughly 3 times that at Lijin. If this is correct, how is this factored into the analysis.

Finally, it’s not clear to me what value of u^* you are using, since it changes with discharge. You may have stated this already, but this dependency of $u^* = f(Q)$ needs to be explored.

→We thank the reviewer for such careful considerations. The short answer is that we did not assume a constant u^* between pre-dam and post-dam conditions, but instead derived the trend (i.e., suspension decreases as channel incises and the bed coarsens) based on the observations that: (1) the grain size of channel bed is usually the first responder to a fluvial disturbance; therefore, the bed coarsens and is followed closely by bedform change, both of which occur more rapidly compared to bed incision that drives a change in u^* (see fig. R7); and (2) in the controlling parameter u^*/v_s , the settling velocity v_s is much more sensitive to grain-size changes (fig. R8) than u^* is to bed incision and slope adaptation ($u^* = (gHS)^{0.5}$).

When bed incision occurs, the general trend is that H increases and S decreases, so the overall shear velocity increase is not as significant (fig. R9). In Table S3, we summarize the observed u^* , v_s , H for the non-impacted and impacted reaches of the LYR, and also note that while the settling velocity increases by a factor of 4 to 6 after damming, the increase in u^* is less than a factor of 2. This can be further evaluated by the hydraulic model in fig. R9, where there is an increase in shear velocity post-damming, but the settling velocity changes much more than shear velocity, thus lowering the suspension number and increasing resistance.

The slower and weaker change of u^* compared to v_s as a consequence of damming is the primary reason why we predict that u^*/v_s will decrease as the bed coarsens. Regarding settling velocity, in the range of silt and sand sediment, there is a power law relation with grain size, from D^2 gradually changing to $D^{0.5}$, with finer material more sensitive to grain size adjustments (i.e., fig. R8).

Lastly, as also inquired by R2, we have developed relations between discharge, shear stress, water depth/stage and velocity based on the hydraulic resistance theory and field observations of channel cross-sections (e.g. fig. R9). Since no data were used to calibrate these relations (e.g. Q_w-Z) beforehand, the results based on the basic hydraulics relation and cross-section constitute predictions that are directly validated by the field data (fig. 5B and fig. S6). We have added these results to the revised manuscript.

Figure R7. Spatial and temporal scales of channel response variables in alluvial rivers. (Fig. 32.1 of Buffington (2012) partially after Knighton (1998)).

Figure R8. Settling velocity diagram based on Ferguson & Church (2004) (courtesy: Dr. Z. Sylvester at <https://hinderedsettling.com/2013/08/09/grain-settling-python/>)

Figure R9. Change in shear velocity, suspension number, resistance coefficient and flow discharge versus flow stage elevation from the thalweg. Although significant post-dam incision leads to an increase in shear velocity (surpassing pre-dam value) the suspension number is much more affected by the offset in the grain size (i.e., pre-dam (1981): 90 microns, versus post-dam (2015): 300 microns). All the plots are based on field data and hydraulic theory, and C_f is calculated based on the proposed relation; there are no tunable parameters to calibrate the relation to data. The same computations were conducted based on 20-yr's of cross-section data, for predicting Q_w - Z data.

154-156: I find this sentence a little confusing. As I understand it, you are primarily using the suspension number as a measure of flow intensity, not because it describes the underlying mechanism resulting in formation of dunes or plane beds. Thus “attributing” the formation of dunes to the suspension number seems off. The Froude number is, in my view, more closely tied to the mechanism of dune formation, since it describes free-surface effects, and is traditionally the basis on which flow and bedform are discriminated (e.g., Simon and Richardson, etc.). But of course, the Froude number refers only to flow dynamics and doesn't include sediment size at all, the latter being central to your argument.

→Indeed, Fr is a major discriminator for dunes, particularly for relatively steep streams, including gravel and/or sand-bedded systems, where Fr of the flow can be as high as >0.8 . It is apparent, however, that for lowland, fine-grained rivers (the focus of our study), Fr is not as high nor particularly variable between discharge conditions. For instance, most of flow cases in LYR have values of Fr less 0.5, which, according to the Fr phase diagram, is associated with dunes of typical size. Nevertheless, we show that at Lijin, the bed is nearly flat, maintaining irregular bedforms. In this case, we argue that Fr is thus not a major controlling factor on the bed state transition in LYR. Instead, grain size controls the bed state, as discussed in the references listed in the above responses.

157: I consider an Fr of 0.7-0.8 to be relatively high. We've seen standing waves beginning to form in sand-bed channels with Fr in this range. In fact, the paper you cite by Naqshband and others (2014; Ref.

37) defines a range of “low” Froude numbers as 0.05–0.32 while “high” Froude numbers range 0.32–0.84.

→In Naqshband et al. (2014) (data shown in Fig. R5, together with new ones), the dominance of the suspension number on bedform geometry is quite clear, in particular for the low Fr number range, but even for the high Fr number range (up to 0.84). For the latter case, bedform shape and state transition can still be well represented by the suspension number. We constrain our research targets to lowland, fine-grained rivers, where the Fr is relatively low, and so the suspension number is a dominating variable.

159: I agree with the point of this sentence and would argue that just showing where the two sites plot on Fig. 2 only suggests or predicts change in dune height; it does not provide evidence of “substantial change”. The latter needs direct observation to confirm, as you suggest.

→Yes, we did show direct observations in fig.3.

162-167: I found the data presented in Fig. 3 problematic, and less convincing than I had hoped. To begin with, the scales of the two reaches are completely different, and it is hard to determine whether the absence of clear dunes at Lijin is real or a consequence of the resolution of the imagery/measurement.

→The vertical resolution of multibeam echo sounder data, when combined with RTK, is millimeters to centimeters for both vertical and horizontal positions (depending on the boat navigation speed and satellite coverage). This is sufficient to capture bedform size at Lijin (i.e., height: ~16- 22 cm) and at Huayuankou (height: ~70-90 cm). As to the planform coverage, we show two multibeam maps from Lijin in fig. 3b and fig. S3a, the length of which are ~ 2.5 km and ~ 1.5 km, with widths ~7 m and ~ 42 m, respectively. Both maps illustrate the low-relief character of the bed topography at Lijin, particularly compared to Huayuankou (i.e., long profiles of fig. 3c and fig. S3 d&e).

The reason for the stretched appearance of the bed profile in fig. 3b and S3a is that the low relief of the dunes necessitates showing them over a very long profile, particularly when compared to the more typically sized dunes that exist at Huayuankou, which can be shown using a relatively short horizontal scale. Additionally, the shallow water depth at Lijin limited the extent of channel area that could be surveyed (as explained in the manuscript). Nevertheless, the data for the 1.5 and 2.5 km long bed survey profiles (fig. 3b and fig. S3a) are sufficient to show clearly the nature of the observed low-relief bedforms.

Could you even detect bedforms if they were present at Lijin using the imagery you show? At a minimum, a clear discussion of the likely error associated with the coarser resolution seems warranted.

→We appreciate the reminder and have now discussed potential uncertainties as related to the statistics of bedform geometry in the Discussion section. We agree that such uncertainties associated with bedform geometry could be important, and this is the reason why we show bedform size data with error bars (fig. 3b). Still, we do not think that there is a major concern with survey resolution as mentioned above. While more data is always desired, in its absence, we argue that uncertainty and variability should be described fully, because the variabilities and uncertainties are inherent with the shape of natural bedforms as has been recounted in a recent study documenting the variability of dune field shapes for the world’s large rivers (Cisneros et al. 2020). We have updated the text of the Discussion to reflect these important points.

I had great difficulty orienting where the data was actually taken from in each reach using the insets and profile lines. Do you interpret the convex up long profile at Lijin as reflecting a long wavelength, low

amplitude form (your reference to the L/H ratio suggests this), yet the B-B' inset shows a wide range of wavelengths and amplitudes? You might have seen a closer resemblance between the two reaches had you chosen a different section to highlight. This figure shows the crux of your argument and needs to be much more compelling than it is.

→We have now clarified the locations where the long profiles are chosen. In this regard, the wavelength-to-height ratio was calculated and averaged, based on handpicked bedforms. The crests and troughs of the bedforms were first identified, and the elevation difference between the adjacent peaks and troughs were sampled to represent bedform height, and the streamwise difference in adjacent troughs were sampled to quantify bedform wavelength. L_b/H_b are calculated for each bedform, and the range of L_b/H_b ratios are shown. We agree with the reviewer that the long profiles have uncertainties, which is why showing the statistics of the data is crucial. Meanwhile, we also want to emphasize that the important features we wish to document are the long wavelengths and low heights of the irregular bedforms. Our analyses serve this purpose well.

196-197: See note for lines 638-9...I don't think your data actually extends this relationship

→Revised.

202-203: It is somewhat surprising to me that there is no relation between C_f and discharge (lines 191-192). If form drag due to bedforms is a primary contributor to overall resistance, one would expect that this resistance term would vary as a function of relative submergence which is directly proportional to discharge. Why doesn't it, either for the pre- or post-dam case when presumably, the shape of the bedforms is changing?

→We are sorry that we were unclear here regarding the relation between C_f and discharge. We also expected, just as the reviewer, that C_f varies with discharge, because discharge variation may set a different shear velocity, and thus a different suspension number, bedform size, and resistance. Due to the sparsity of data, the trend between C_f and q is not statistically significant in either the pre- or post-dam sub-dataset. We have revised the presentation of the figure, which now shows a box plot. As mentioned in the manuscript, the trend became very clear as we put all the data together and compared them to an empirical curve.

205-206: Remove passive voice

→Done

231-243: I find Figure 5 compelling, and yet it raises question. Considering that you find no relationship between C_f and specific discharge in the data, as discussed above, how can you then argue for a discharge dependency here? Unless I missed something, this strikes me as implausible, or at least requiring explanation.

→We appreciate this comment. As mentioned above, the trend between C_f and discharge was not statically significant, as we plot data only from pre- or post-dam, which is probably due to the lack of data in either case. But when plotted together on the empirical curve, a trend clearly emerges. We have revised the text to state this more clearly.

Moreover, you argue (239-240) for a coupling between bed incision and bed coarsening. I don't see any proposed mechanism to explain this, which gets at a larger point: the lack of a physical mechanism linking coarsening and resistance through bedform development. I think you've made a reasonable case for a correlation between the two, but there's been no discussion (up to now) as to how? This question of how applies to both bed coarsening and its link with bedform development. Is coarsening the consequence of armor development (which seems quite difficult to visualize in such fine-grained rivers)? Or is depletion of the fine fraction or...? And how

does this lead to larger or steeper bedforms? These questions need to be introduced and explored.

→We thank the reviewer for this important feedback. We have sought in our revisions to state clearly how this process emerges. Indeed, this was also addressed in our comments to R1&R2 above, where as we mention, bed coarsening originates from selective entrainment of sediment with different grain sizes. Finer sediment is preferentially entrained and transported, and as the sediment supply is depleted, the median grain size coarsens over time. Again, as mentioned above, this has been well described in recent papers, specifically, Naito et al. (2019) and An et al. (2021), where the sediment flux of finer sediment in a sediment mixture transport is shown to be much greater than that of coarser sediment.

258-265: I find this entire section confusing. First, what do you mean by “transport stage parameter” (line 259)? I’m also not clear what problem this enumerated list is targeted at, i.e., what exactly are all these refinements intended to help illuminate?

→We apologize for the confusion. The existing relations between bedform size and hydraulic parameters, e.g. Fr, suspension number, or the ratio of shear stress to the critical shear stress etc., still possess large uncertainties, including the proposed relation in this manuscript, although its performance surpasses existing ones. In order to further reduce uncertainties of the predictive relation, and increase its accuracy, we proposed several possible physical mechanisms that should improve the existing relations. We have elaborated on these points in the revised manuscript.

299: I suggest using the word “routing” instead of “retaining” flood water.

→Revised.

360-362: Here or in S2, briefly describe methods of sampling bed material, i.e., how was coring done?

→We deployed a tripod from the survey boat, and from floating bridges when they are nearby, and collected the sediment via vibracoring. We have elaborated on this methodological step in the revised manuscript.

381-384: Reword for clarity. Also, is stratigraphy measured in the cores, or did you just assume that the coarsest fraction represented the surface layer? How deep is the surface layer?

→Revised. The length of the cores ranged between 0.7m to 2m. We sampled sediment every 10 cm of each core; however, only the surface sediment samples from thalweg are used in this study (thus the surface sediment is presumed to represent the surface sediment of the core to a depth of 10 cm). We did not assume the coarsest fraction represented the surface layer, but instead used the sediment sample for the thalweg, which is usually the coarsest among the surface samples from the cores at each cross-section. Again, we have revised the Methods to reflect this information.

421: What do you mean by “normal flow and sediment transport equilibrium”?

→We mean to say that the flow is, on average, steady and uniform and there is little net erosion or deposition, so the sediment transport regime is close to equilibrium. We have now clarified this in the revised manuscript.

638-39: Logic unclear; since you are using independently measured bedform height as a way of testing whether the predicted empirical relation is borne out, you can’t really say that it allows you to extend the data field for that empirical relation as well.

→Revised.

References cited

Sternberg, R. W., Berhane, I., & Ogston, A. S. (1999). Measurement of size and settling velocity of suspended aggregates on the northern California continental shelf. *Marine Geology*, 154(1-4), 43-53.

References cited only in the Response Letter:

- B. Andreotti, P. Claudin, O. Devauchelle, O. Durán, A. Fourrière, Bedforms in a turbulent stream: ripples, chevrons and antidunes. *Journal of Fluid Mechanics* **690**, 94-128 (2012).
- J. Best, The fluid dynamics of river dunes: A review and some future research directions. *Journal of Geophysical Research: Earth Surface* **110**, F04S02, doi:10.1029/2004JF000218 (2005).
- J. M. Buffington. *Changes in channel morphology over human time scales* [Chapter 32]. In: M. Church, P. M. Biron, A. G. Roy, eds. *Gravel-Bed Rivers: Processes, Tools, Environments*. Chichester, UK: Wiley, 435-463 (2012).
- R. Ferguson, M. Church, A simple universal equation for grain settling velocity. *Journal of Sedimentary Research* **74**, 933-937 (2004).
- D. Knighton, *Fluvial forms and processes: a new perspective*. (Routledge, 2014).
- S. Naqshband, A. J. F. Hoitink, Scale-Dependent Evanescence of River Dunes During Discharge Extremes. *Geophysical Research Letters* **47**, e2019GL085902 (2020).
- J.-I. Van Den Berg, A. Van Gelder, A new bedform stability diagram, with emphasis on the transition of ripples to plane bed in flows over fine sand and silt. *Alluvial Sedimentation (Special Publication 17 of the IAS)* **66**, 11 (1993).
- Petersen-Overleir, A. (2006) Modelling stage—discharge relationships affected by hysteresis using the Jones formula and nonlinear regression. *Hydrological Sci J*, **51**, 365–388.
- Yang, S.-Q. and Kelly, S. (2015) The Use of Coastal Reservoirs and SPP Strategy to Provide Sufficient High Quality Water to Coastal Communities. *Journal of Geoscience and Environment Protection*, **3**, 80-92. <http://dx.doi.org/10.4236/gep.2015.35010>.
- Yellow River Conservancy Commission (YRCC). *Integrated Planning of the Yellow River Basin (2012-2030)*. (The Yellow River Water Conservancy Press, 2013) (in Chinese)

Reviewers' Comments:

Reviewer #2:

Remarks to the Author:

The paper has been much improved by enriching the Analysis and Discussions. I also feel that the authors did a good job in the Methods description. The reviewer recommends the acceptance of this manuscript for publication.

Reviewer #3:

Remarks to the Author:

The authors have sincerely addressed the concerns of all three reviewers and the manuscript has been significantly strengthened as a result. In particular, I appreciate the depth and thoughtfulness which the authors brought to addressing my concerns, which were laid out in depth in my previous review. I think the manuscript can move forward in its present form. Gordon Grant

Responses to the second-round comments and reviews on “Amplification of downstream flood stage due to damming of fine-grained rivers” by Ma et al.

We were very encouraged that, after the second round of reviews, both reviewers were satisfied with our revisions and recommended publication in its present form.

Reviewer #2

The paper has been much improved by enriching the Analysis and Discussions. I also feel that the authors did a good job in the Methods description. The reviewer recommends the acceptance of this manuscript for publication.

Reviewer #3

The authors have sincerely addressed the concerns of all three reviewers and the manuscript has been significantly strengthened as a result. In particular, I appreciate the depth and thoughtfulness which the authors brought to addressing my concerns, which were laid out in depth in my previous review. I think the manuscript can move forward in its present form. -Gordon Grant